# The Initialization Determines Whether In-Context Learning Is Gradient Descent

**Shifeng Xie**                                                    *shifeng.xie@telecom-paris.fr*
*Telecom Paris*
*Institut Polytechnique de Paris*
*France*

**Rui Yuan**                                                            *yy42606r@gmail.com*
*Lexsi Labs, Paris*
*France*

**Simone Rossi**                                                    *simone.rossi@eurecom.fr*
*EURECOM*
*France*

**Thomas Hannagan**                                        *thomas.hannagan@stellantis.com*
*Stellantis*
*France*

**Reviewed on OpenReview:** *https://openreview.net/forum?id=fvqSKLDtJi*

## Abstract

In-context learning (ICL) in large language models (LLMs) is a striking phenomenon, yet its underlying mechanisms remain only partially understood. Previous work connects linear self-attention (LSA) to gradient descent (GD), this connection has primarily been established under simplified conditions with zero-mean Gaussian priors and zero initialization for GD. However, subsequent studies have challenged this simplified view by highlighting its overly restrictive assumptions, demonstrating instead that under conditions such as multi-layer or nonlinear attention, self-attention performs optimization-like inference, akin to but distinct from GD. We investigate how multi-head LSA approximates GD under more realistic conditions—specifically when incorporating non-zero Gaussian prior means in linear regression formulations of ICL. We first extend multi-head LSA embedding matrix by introducing an initial estimation of the query, referred to as the *initial guess*. We prove an upper bound on the number of heads needed for ICL linear regression setup. Our experiments confirm this result and further observe that a performance gap between one-step GD and multi-head LSA persists. To address this gap, we introduce $y_q$-LSA, a simple generalization of single-head LSA with a trainable initial guess $y_q$. We theoretically establish the capabilities of $y_q$-LSA and provide experimental validation on linear regression tasks, thereby extending the theory that bridges ICL and GD. Finally, inspired by our findings in the case of linear regression, we consider widespread LLMs augmented with initial guess capabilities, and show that their performance is improved on a semantic similarity task.

## 1 Introduction

Large language models (LLMs) exhibit the interesting phenomenon of in-context learning (ICL), whereby models adapt to new tasks from a few input-label pairs presented in the context, without parameter updates (Brown et al., 2020; Dong et al., 2024). This capability has motivated extensive efforts to clarify the underlying mechanisms. A prominent line of work interprets ICL in simplified linear regression settings as implicitly

performing gradient descent (GD) within a forward pass of linear self-attention (LSA) (Garg et al., 2022; Von Oswald et al., 2023).

However, this equivalence has mostly been established under restrictive assumptions, notably zero-mean Gaussian priors for regression weights and zero initialization for GD. Recent work indicates that these conditions are fragile: Zhang et al. (2024b) showed that introducing a non-zero mean prior produces a persistent gap between LSA and GD, undermining previous guarantees (see Fig. 1). From a modeling perspective, the assumption of a zero-mean prior is quite restrictive: it corresponds to a learner that believes all task-specific regression weights are centered around the origin in parameter space. In realistic pre-training regimes, transformers are exposed to broad data distributions and acquire a shared bias across tasks, which is naturally captured by a non-zero prior mean. These findings raise a fundamental question: *"under what conditions can LSA faithfully recover GD, and when does it fundamentally fail?"*

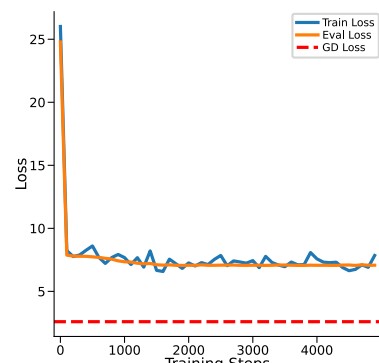

Figure 1: Training and evaluation loss curves of LSA with a non-zero prior mean. The dashed red line denotes the baseline loss achieved by one-step GD.

In this paper, we revisit the ICL-GD connection under more realistic assumptions, explicitly incorporating non-zero prior means and systematically analyzing the role of attention heads and initialization. Our study reveals that the decisive factor is the initialization of the query's prediction, which we term the *initial guess* $y_q$. Misalignment between $y_q$ and the prior induces a persistent gap that cannot be resolved by simply increasing the number of heads. Motivated by this observation, we propose $y_q$-LSA, an architectural extension that incorporates a trainable initialization mechanism, thereby restoring equivalence with GD even in the non-zero mean setting.

Our analysis is closely related to the concurrent work of Zhang et al. (2024b), who study a linear transformer block (LTB) obtained by composing an LSA layer with linear multi-layer perceptron. In the non-zero mean setting, they show that an optimally trained LTB implements a preconditioned, near-Newton update with a learnable initialization. In contrast, we ask to what extent the ICL to GD correspondence can be recovered with LSA architectures. From this viewpoint, $y_q$-LSA can be seen as a minimal extension that acts only on the input-side initialization, isolating the role of the query's initial guess without adding extra layers or a large number of additional parameters.

**Contributions.** This work makes the following contributions:

1. We prove that when regression weights have a non-zero mean, multi-head LSA cannot in general replicate one-step GD, even with arbitrarily many heads, establishing a fundamental limitation of the ICL-GD correspondence.

2. We show that the query initialization $y_q$ is the decisive factor: misalignment induces a persistent gap, while correcting $y_q$ suffices to recover GD even with a single head.

3. We propose $y_q$-LSA, an extension of LSA with a trainable initialization vector, and demonstrate both theoretically and empirically that it restores equivalence with GD in the non-zero mean setting.

4. We provide proof-of-concept experiments showing that introducing explicit initial guesses improves ICL performance in LLMs, thereby linking our theoretical results with practical prompting strategies.

**Scope.** Our analysis focuses on linear regression with linear self-attention, a simplified but analytically tractable setting. Within this framework, we identify precise conditions under which LSA diverges from gradient descent and propose $y_q$-LSA as a principled correction. These results provide a foundation for extending analysis to richer transformer architectures, including softmax attention and multi-layer models.

## 1.1 Related Work

Theoretical studies on ICL have analyzed its mechanisms to understand how LLMs effectively learn from contextual examples (Brown et al., 2020). ICL can be framed as an implicit Bayesian process where the model performs posterior inference over a latent task structure based on contextual examples, performing a form of posterior updating (Xie et al., 2022; Falck et al., 2024; Panwar et al., 2024; Ye et al., 2024). Alternatively, a more recent perspective suggests that ICL in transformers is akin to gradient-based optimization occurring within their forward pass. Von Oswald et al. (2023) demonstrate that self-attention layers can approximate gradient descent by constructing task-specific updates to token representations. They provide a mechanistic explanation by showing how optimized transformers can implement gradient descent dynamics with a given learning rate (Rossi et al., 2024; Zhang et al., 2025). While this work provides a new perspective on ICL, it limits the analysis to simple regression tasks and it simplifies the transformer architecture by considering a single-head self-attention layer without applying the $\mathsf{sfmx}(\cdot)$ function on the attention weights (also known as linear attention). Ahn et al. (2023) extend the work of Von Oswald et al. (2023) by showing how the in-context dynamics can learn to implement *preconditioned* gradient descent, where the preconditioner is implicitly optimized during pretraining. More recently, Mahankali et al. (2024) prove that a single self-attention layer converges to the *global* minimum of the squared error loss. Zhang et al. (2024b); Wang et al. (2025) also analyze a more complex transformer architecture with a (linear) multi-layer perceptron (MLP) or softmax after the linear self-attention layer, showing the importance of such block when pretraining for more complex tasks. In a related direction, Cheng et al. (2024) show that transformers can implement functional gradient descent to learn non-linear functions in context, further strengthening the view of ICL as gradient-based optimization.

Recent works have also raised important critiques of the ICL to GD hypothesis, questioning both its theoretical assumptions and empirical applicability. For example, Shen et al. (2023; 2024) point out that many theoretical results—such as those in Von Oswald et al. (2023)—rely on overly simplified settings, including linearized attention mechanisms, handcrafted weights, or order-invariant assumptions not satisfied in real models. Giannou et al. (2024); Fu et al. (2024) demonstrated that in a multi-layer self-attention setting, the internal iterations of the Transformer conform more closely to the second-order convergence speed of Newton's Method. Therefore, the interpretation of ICL needs to be examined under more realistic assumptions.

In this work, we extend the above lines of research by emphasizing more realistic priors, specifically, non-zero prior means. While Zhang et al. (2024a); Mahdavi et al. (2024) explore broader prior distributions by analyzing covariate structures or modify the distribution of input feature, our focus instead lies on the interplay between a non-zero prior mean and the capacity of LSA to emulate GD. We note that while Ahn et al. (2023); Mahankali et al. (2024); Zhang et al. (2024b) provide compelling theoretical analyses, their work does not include experimental validations. In doing so, our study builds upon and generalizes the prior-zero analyses found in Von Oswald et al. (2023); Ahn et al. (2023), illuminating new challenges and insights that arise when priors deviate from zero, both theoretically and empirically.

## 2 Preliminaries

We use $\mathbf{x} \in \mathbb{R}^d$ and $y \in \mathbb{R}$ to denote a feature vector and its label, respectively. We consider a fixed number of context examples, denoted by $C > 0$. We denote the context examples as $(\boldsymbol{X}, \mathbf{y}) \in \mathbb{R}^{C \times d} \times \mathbb{R}^C$, where each row represents a context example, denoted by $(\mathbf{x}_i^\top, y_i)$, $i \in [C]$. That is,

$$\boldsymbol{X} \overset{\text{def}}{=} \begin{bmatrix} \mathbf{x}_1^\top \\ \vdots \\ \mathbf{x}_C^\top \end{bmatrix} \in \mathbb{R}^{C \times d} \quad \text{and} \quad \mathbf{y} \overset{\text{def}}{=} \begin{bmatrix} y_1 \\ \vdots \\ y_C \end{bmatrix} \in \mathbb{R}^C. \tag{1}$$

To formalize an in-context learning (ICL) problem, the input of a model is an *embedding matrix* given by

$$\boldsymbol{E} \overset{\text{def}}{=} \begin{bmatrix} \boldsymbol{X}^\top & \mathbf{x}_q \\ \mathbf{y}^\top & y_q \end{bmatrix} \in \mathbb{R}^{(d+1) \times (C+1)}, \tag{2}$$

where $\mathbf{x}_q \in \mathbb{R}^d$ is a new query input and $y_q \in \mathbb{R}$ is an *initial guess* of the prediction for the query $\mathbf{x}_q$. The model's output corresponds to a prediction of $y \in \mathbb{R}$. Notice that the embedding matrix in equation 2 is a

slight extension to the commonly used embedding matrix, e.g. presented in Von Oswald et al. (2023), where $y_q$ is set to be zero by default. Its interpretation will be clearer in the next two sections.

**Linear regression tasks.** We formalize the linear regression tasks as follows. Assume that $(\boldsymbol{X}, \mathbf{y}, \mathbf{x}_q, y)$ are generated by:

- First, a task parameter is independently generated by $\widehat{\mathbf{w}} \sim \mathcal{N}(\mathbf{w}_\star, \boldsymbol{I}_d)$, where $\mathcal{N}(\mathbf{w}_\star, \boldsymbol{I}_d)$ is the *prior*, and $\mathbf{w}_\star$ is called the *prior mean*.

- The feature vectors are independently generated by $\mathbf{x}_q, \mathbf{x}_1, \dots \mathbf{x}_C \overset{\text{i.i.d.}}{\sim} \mathcal{N}(0, \boldsymbol{I}_d)$.

- Then, the labels are generated by $y = \langle \widehat{\mathbf{w}}, \mathbf{x}_q \rangle$, and $y_i = \langle \widehat{\mathbf{w}}, \mathbf{x}_i \rangle$, $i \in [C]$, with no noise.

Here, $\mathbf{w}_\star \in \mathbb{R}^d$ is fixed but unknown and governs the data distribution.

**A linear self-attention.** We consider a *linear self-attention* (LSA) defined as

$$f_{\mathsf{LSA}} : \mathbb{R}^{(d+1) \times (C+1)} \to \mathbb{R},$$
$$\boldsymbol{E} \mapsto \left[ \boldsymbol{E} + \tfrac{1}{C} \mathbf{W}^P \mathbf{W}^V \boldsymbol{E} \mathbf{W}^M (\boldsymbol{E}^\top (\mathbf{W}^K)^\top \mathbf{W}^Q \boldsymbol{E}) \right]_{-1,-1}, \tag{3}$$

where $\mathbf{W}^K, \mathbf{W}^Q, \mathbf{W}^P, \mathbf{W}^V \in \mathbb{R}^{(d+1) \times (d+1)}$ are trainable parameters, $[\,\cdot\,]_{-1,-1}$ refers to the bottom right entry of a matrix, and $\mathbf{W}^M \overset{\text{def}}{=} \begin{bmatrix} \boldsymbol{I}_C & 0 \\ 0 & 0 \end{bmatrix}$ is a mask matrix. Our linearized self-attention removes softmax, LayerNorm, and nonlinear activations. Consequently, the update is an affine function of low-order context aggregates (e.g., $X^\top X$, $X^\top y$), which enables closed-form analysis of initialization effects while preserving the in-context learning setup.

**ICL risk.** We measure the ICL risk of a model $f$ by the mean squared error,

$$\mathcal{R}(f) \overset{\text{def}}{=} \mathbb{E}[(f(\boldsymbol{E}) - y)^2], \tag{4}$$

where the input $\boldsymbol{E}$ is defined in equation 2 and the expectation is over $\boldsymbol{E}$ (equivalent to over $\boldsymbol{X}$, $\mathbf{y}$, and $\mathbf{x}_q$) and $y$. The performance of different models are characterized by the ICL risk.

## 3 Multi-Head Linear Self-Attention

In order to improve the performance of linear self-attention (LSA), we consider the multi-head extension. Let $H \in \mathbb{N}$ be the number of heads. Similar to equation 3, we define the output of each transformer head as

$$\text{head}_h(\boldsymbol{E}) \overset{\text{def}}{=} \frac{1}{C} \mathbf{W}_h^P \mathbf{W}_h^V \boldsymbol{E} \mathbf{W}^M \left( \boldsymbol{E}^\top (\mathbf{W}_h^K)^\top \mathbf{W}_h^Q \boldsymbol{E} \right), \quad h \in [H], \tag{5}$$

where $\mathbf{W}_h^K, \mathbf{W}_h^Q, \mathbf{W}_h^P$ and $\mathbf{W}_h^V$ are trainable parameters specific to the $h$-th head. The multi-head LSA function is defined as

$$f_{\mathsf{H-LSA}}(\boldsymbol{E}) \overset{\text{def}}{=} \left[ \boldsymbol{E} + \sum\nolimits_{h=1}^{H} \text{head}_h(\boldsymbol{E}) \right]_{-1,-1}. \tag{6}$$

Standard multi-head attention concatenates head outputs and applies a linear projection $W^O$. Algebraically, $\text{Concat}(\text{head}_1, \dots, \text{head}_H)W^O$ equals $\sum_{h=1}^{H} W_h^P \text{head}_h$ after absorbing $W^O$ into per-head projections $\{W_h^P\}$. We therefore use a sum without loss of generality, keep the model dimension $(d+1)$, and retain per-head contribution after reparameterization.

We emphasize that both the single-head LSA $f_{\mathsf{LSA}}$ and the multi-head LSA $f_{\mathsf{H-LSA}}$ share a common structural property: the bottom-right entry of the output matrix corresponds to the prediction for the query point $x_q$, which can be interpreted as an *initial guess* $y_q$ refined by an attention-based update. In the special case of linear regression with zero prior mean, i.e., $\mathbf{w}_\star = 0$, the choice $y_q = 0$ introduces a non-trivial prior for the

initial guess, as already observed by Von Oswald et al. (2023). The empirical role of this initial guess in the multi-head setting will be further analyzed in Section 5.1.3.

We denote by

$$\mathcal{F}_{H-\mathsf{LSA}} \stackrel{\text{def}}{=} \left\{ f_{\mathsf{H-LSA}} \;\middle|\; \left\{ \mathbf{W}_h^K, \mathbf{W}_h^Q, \mathbf{W}_h^V, \mathbf{W}_h^P \right\}_{h=1}^H \right\}$$

the hypothesis class associated with multi-head LSA models with $H$ heads. Our first theoretical result establishes an invariance of the optimal in-context learning risk with respect to the number of heads once it exceeds the feature dimension.

**Theorem 1.** *Let $d \in \mathbb{N}$, and consider the hypothesis classes $\mathcal{F}_{(d+1)-\mathsf{LSA}}$ and $\mathcal{F}_{(d+2)-\mathsf{LSA}}$ corresponding to multi-head LSA models with $H = d+1$ and $H = d+2$ attention heads, respectively. Then*

$$\inf_{f \in \mathcal{F}_{(d+1)-\mathsf{LSA}}} \mathcal{R}(f) \;=\; \inf_{f \in \mathcal{F}_{(d+2)-\mathsf{LSA}}} \mathcal{R}(f),$$

*where $\mathcal{R}(f)$ is the ICL risk defined in Eq. (4).*

While the full proof of Theorem 1 is provided in Appendix A.1, we outline the key intuition here. Each attention head contributes a rank-one update to a set of $(d+1)$ matrices that fully describe the model. Collectively, these matrices live in a space of dimension $(d+1)^3$. A single head provides $(d+1)(d+2)$ degrees of freedom, so once the number of heads reaches $d+1$, the parameter space already has enough capacity to span the entire target space. In fact, with $d+1$ heads one can explicitly construct any target configuration, which means the model is already maximally expressive. Since adding further heads simply amounts to appending zero-contributing heads, the hypothesis class does not grow beyond $d+1$ heads, and the achievable risk remains unchanged. In Section 5, we provide empirical evidence supporting this theoretical result across a variety of model configurations.

**Relation to concurrent work.** Theorem 1 is a capacity statement for linear self-attention (LSA): once the number of heads reaches $H = d+1$, the hypothesis class and the attainable ICL risk no longer improve by adding heads. This contrasts with results for softmax attention, where (Cui et al., 2024) give exact risk formulas for single/multi-head ICL and show that as the number of in-context examples $C$ grows, both risks scale as $O(1/C)$ but multi-head achieves a smaller multiplicative constant when the embedding dimension is large—an improvement in performance constants rather than capacity. Complementarily, (Chen et al., 2024) study trained multi-layer transformers and find that multiple heads matter primarily in the first layer, proposing a preprocess-then-optimize mechanism; their conclusions concern learned utilization patterns (with softmax and multi-layer architectures), whereas Theorem 1 isolates an expressivity saturation specific to single-layer LSA.

Next, we explore the convergence of multi-head LSA. Inspired by the analysis of Ahn et al. (2023), we analyze the stationary point of the ICL risk for multi-head LSA functions.

**Theorem 2.** *Let $H \in \mathbb{N}$ and consider the hypothesis class $\mathcal{F}_{H-\mathsf{LSA}}$ of multi-head LSA models with context size $C \to \infty$. Then the in-context learning risk $\mathcal{R}(f)$ admits no non-trivial stationary point in parameter space. More precisely,*

$$\nabla \mathcal{R}(f) \;\neq\; 0 \quad \text{for all } f \in \mathcal{F}_{H-\mathsf{LSA}}$$

*for every choice of parameters $\{\mathbf{W}_h^K, \mathbf{W}_h^Q, \mathbf{W}_h^V, \mathbf{W}_h^P\}_{h=1}^H$, except in the case where the prior mean vector vanishes, $\mathbf{w}_\star = 0$.*

Theorem 2 states that when the context size $C \to \infty$, the gradient of the multi-head LSA's ICL risk $\mathcal{R}(f_{\mathsf{H-LSA}})$ remains non-zero for the entire parameters space as long as $\mathbf{w}_\star \neq 0$. This result highlights a fundamental limitation of multi-head LSA under non-zero priors: no choice of weights $\mathbf{W}_h^K, \mathbf{W}_h^Q, \mathbf{W}_h^P$ and $\mathbf{W}_h^V$ with $h \in [H]$ can minimize the ICL risk in the infinite-context limit. See Appendix A.3 for a detailed discussion of how the results change when the context size $C$ is finite.

**Relation to concurrent work.** Although previous works such as Ahn et al. (2023) and Mahankali et al. (2024) provide analytical solutions corresponding to stationary points of the ICL risk, these results are derived under the assumption that the prior mean $\boldsymbol{w}_\star = 0$. In this special case, the gradient of the ICL risk can

vanish, allowing the existence of a stationary point. Our analysis generalizes this observation: we prove that when $\boldsymbol{w}_\star \neq 0$, the gradient of the ICL risk remains strictly non-zero for all weights as context size $C \to \infty$, thus precluding the existence of stationary points. We adopt $C \to \infty$ as an asymptotic approach, as done by Zhang et al. (2024a); Huang & Ge (2024). Our analysis targets the asymptotic regime $C \to \infty$, where finite-sample correlation terms vanish and the gradient remains strictly non-zero for $\mathbf{w}_\star \neq 0$, hence no non-trivial stationary points exist. For fixed, finite $C$, an additional finite-sample correction—decaying inversely with $C$—can partially cancel the leading gradient, producing apparent stationary points or plateaus in practice. As $C$ grows, these effects fade and the behavior converges to the asymptotic prediction, matching our experiments.

Finally, even though such a stationary point exists with finite context size, we still cannot imply that the stationary point is the global optimum, as the ICL risk of multi-head LSA $\mathcal{R}(f_{\mathsf{H-LSA}})$ is not convex, presented in the following lemma.

**Lemma 1.** *For any $H \in \mathbb{N}$, the in-context learning risk*

$$\mathcal{R}(f), \qquad f \in \mathcal{F}_{H-\mathsf{LSA}},$$

*is not convex in the parameters $\{\mathbf{W}_h^K, \mathbf{W}_h^Q, \mathbf{W}_h^V, \mathbf{W}_h^P\}_{h=1}^H$.*

Because $\mathcal{R}(f_{\mathsf{H-LSA}})$ is non-convex, any stationary point that arises, even at finite context sizes, does not guarantee a global optimum. In other words, one may encounter local minima or saddle points that satisfy the stationary condition without minimizing the overall ICL risk.

## 4 $y_q$-Linear Self-Attention

To address the performance gap between one-step GD and multi-head LSA, we introduce $y_q$-LSA, a generalization of single-head LSA.

### 4.1 Formulation of $y_q$-LSA

Our approach builds upon the *GD-transformer* developed by Von Oswald et al. (2023); Rossi et al. (2024), which implements one-step GD in a linear regression setup when the prior mean $\mathbf{w}_\star$ is zero. The original formulation is defined by the weight matrices

$$\mathbf{W}^V = \begin{bmatrix} 0 & 0 \\ \mathbf{w}_\star^\top & -1 \end{bmatrix}, \quad \mathbf{W}^K = \mathbf{W}^Q = \begin{bmatrix} \boldsymbol{I}_d & 0 \\ 0 & 0 \end{bmatrix}, \quad \mathbf{W}^P = -\frac{\eta}{C}\boldsymbol{I}_{d+1}, \tag{7}$$

where $\eta$ represents the GD step size. From the standard LSA formulation equation 3 with the given embedding equation 2, we derive

$$f_{\mathsf{LSA}}(\boldsymbol{E}) = y_q - \frac{\eta}{C}(\mathbf{w}_\star^\top \boldsymbol{X}^\top - \mathbf{y}^\top)\boldsymbol{X}\mathbf{x}_q, \tag{8}$$

where the initial guess $y_q = 0 = \mathbf{w}_\star^\top \mathbf{x}_q$ is fixed for any query $\mathbf{x}_q$, and the prior mean $\mathbf{w}_\star$ is zero. See the derivation of equation 8 in Appendix B for the completeness. Notably, we retain the terms for $y_q$ and $\mathbf{w}_\star$ to facilitate future extension to non-zero scenarios. Rewriting the equation equation 8 with $y_q = \mathbf{w}_\star^\top \mathbf{x}_q$ yields

$$f_{\mathsf{LSA}}(\boldsymbol{E}) = \left(\mathbf{w}_\star - \frac{\eta}{C}\boldsymbol{X}^\top(\boldsymbol{X}\mathbf{w}_\star - \mathbf{y})\right)^\top \mathbf{x}_q. \tag{9}$$

The red term represents the gradient of the least-squares loss in linear regression. Consequently, $f_{\mathsf{LSA}}(\boldsymbol{E})$ becomes equivalent to a linear function $f(\mathbf{x}_q) = \mathbf{w}^\top \mathbf{x}_q$, where $\mathbf{w}$ is the one-step GD update initialized at the prior mean $\mathbf{w}_\star$.

For the more general case with a non-zero prior mean $\mathbf{w}_\star$, we relax the condition on the initial guess $y_q$. By allowing $y_q$ to be a linear function of $x_q$, specifically $y_q = \mathbf{w}_\star^\top \mathbf{x}_q$, we obtain the prediction of the linear regression task with a given query $\mathbf{x}_q$

$$\left(\mathbf{w}_\star - \frac{\eta}{C}\boldsymbol{X}^\top(\boldsymbol{X}\mathbf{w}_\star - \mathbf{y})\right)^\top \mathbf{x}_q, \tag{10}$$

which still implements the one-step GD update. Given this, we can now define $y_q$-LSA.

**Definition 3** ($y_q$-LSA)**.** *We define $y_q$-LSA with a flexible initial guess embedding matrix*

$$\boldsymbol{E}_{\mathbf{w}} \stackrel{def}{=} \begin{bmatrix} \boldsymbol{X}^\top & \mathbf{x}_q \\ \mathbf{y}^\top & y_q \end{bmatrix} \in \mathbb{R}^{(d+1)\times(C+1)}, \quad with \; y_q = \mathbf{w}^\top \mathbf{x}_q, \tag{11}$$

*where $\mathbf{w} \in \mathbb{R}^d$ is a trainable parameter and $y_q$ is the initial guess. The $y_q$-LSA function is defined as*

$$f_{y_q-\mathsf{LSA}}(\boldsymbol{X}, \mathbf{y}, \mathbf{x}_q) \stackrel{def}{=} f_{\mathsf{LSA}}(\boldsymbol{E}_{\mathbf{w}}). \tag{12}$$

The $y_q$-LSA extends the standard LSA by introducing an additional parameter $\mathbf{w}$ in the embedding, enabling better alignment with the query's initial guess. The trainable parameters of $y_q$-LSA now include $\mathbf{W}^K, \mathbf{W}^Q, \mathbf{W}^P, \mathbf{W}^V$ and $\mathbf{w}$, with inputs $\boldsymbol{X}, \mathbf{y}$ and $\mathbf{x}_q$.

### 4.2  Analysis of $y_q$-LSA

Similar to the analysis of multi-head LSA, we first examine the stationary point of $y_q$-LSA.

**Theorem 4.** *For a $y_q$-LSA function in equation 12 with a non-zero prior mean $\mathbf{w}_\star$ and context size $C \to \infty$, the weights $(\mathbf{W}^K, \mathbf{W}^Q, \mathbf{W}^P, \mathbf{W}^V, \mathbf{w}_\star)$ in equation 7 with $\mathbf{w} = \mathbf{w}_\star$ constitute a stationary point of $\mathcal{R}(f_{y_q-\mathsf{LSA}})$.*

Theorem 4 is asymptotic in the context length $C$: when $C \to \infty$, the gradient vanishes at the weights in Eq. (7) with $\mathbf{w} = \mathbf{w}_\star$. For finite $C$, each gradient component differs from its infinite-$C$ value by a correction of order $1/C$. Thus $\mathbf{w} = \mathbf{w}_\star$ behaves as an approximate stationary point whose residual gradient (and the resulting bias) decays as $C$ grows, explaining the small plateaus occasionally observed at finite $C$. Similar to multi-head LSA, we cannot conclusively determine that this stationary point represents the global optimum. This uncertainty comes from the non-convex nature of the $y_q$-LSA ICL risk, as established in the following lemma.

**Relation to concurrent work.** Unlike Ahn et al. (2023)—who show that single-layer LSA attains one-step preconditioned GD under a zero-mean prior—Theorem 4 establishes that with a non-zero prior mean, one-step GD is still recovered without an MLP by introducing a trainable query initialization $y_q = \mathbf{w}^\top \mathbf{x}_q$. In contrast to Zhang et al. (2024b), where an LTB (LSA+MLP) realizes GD-$\beta$/near-Newton via the MLP, our result identifies input-side initialization as the minimal mechanism that closes the ICL–GD gap within LSA.

**Lemma 2.** *The ICL risk of $y_q$-LSA $\mathcal{R}(f_{y_q-\mathsf{LSA}})$ is not convex.*

While the non-convexity prevents a definitive proof of global optimality, our empirical investigations in Section 5.2 suggest an intriguing hypothesis. Notably, we conjecture that the stationary point identified in Theorem 4 may indeed be the global optimum. Empirical evidence indicates that the performance of one-step gradient descent serves as a lower bound for $y_q$-LSA.

An additional noteworthy observation is $y_q$-LSA's relationship to the *linear transformer block* introduced by Zhang et al. (2024b). Unlike $y_q$-LSA, LTB combines LSA with a linear multilayer perceptron (MLP) component. Critically, the global optimum of LTB implements a Newton step rather than one-step gradient descent. This approach fails to bridge the performance gap between one-step GD and single-head LSA and requires significantly more parameters through the additional MLP, in contrast to $y_q$-LSA's more parsimonious approach of introducing a single vector parameter $\mathbf{w}$. See Lemma 3 in Appendix B for more details.

## 5  Experiments

For experiments in Sections 5.1 and 5.2, we focus on a simplified setting where the LSA consists of a single linear self-attention layer without LayerNorm or softmax. We generate linear functions in a 10-dimensional input space ($d = 10$) and provide $C = 10$ context examples per task. We endow the LSA parameters with ICL capability by minimizing the expected ICL risk $\mathbb{E}[(f_\theta(E) - y)^2]$ over random tasks. Each training step is an Adam update of $\{W^Q, W^K, W^V, W^P\}$ (and $\mathbf{w}$ for $y_q$-LSA) using freshly sampled $(\boldsymbol{X}, \mathbf{y}, \mathbf{x}_q, y)$; at test time, no parameter updates are performed. We train for 5000 gradient steps. Further implementation details are provided in Appendix C.1.

### 5.1 Multi-head LSA

#### 5.1.1 Multi-head LSA with Varying Numbers of Heads

We investigate the ICL risk (evaluation loss) of the multi-head LSA under different numbers of attention heads in the setting of a non-zero prior mean and $y_q$ is fixed at zero (details in Table 1). Fig. 2a illustrates the loss curves over the course of training for several head configurations, while Fig. 2b summarizes the final evaluation losses as a function of the number of heads. From these results, we observe that increasing the number of heads up to $d + 1$ (here $d = 10$, see Fig. 2b) substantially enhances the in-context learning capability of multi-head LSA, as reflected by a pronounced reduction in the final evaluation loss.

However, adding more than $d + 1$ heads yields negligible further improvement, indicating a saturation effect beyond this threshold. This confirms our results in

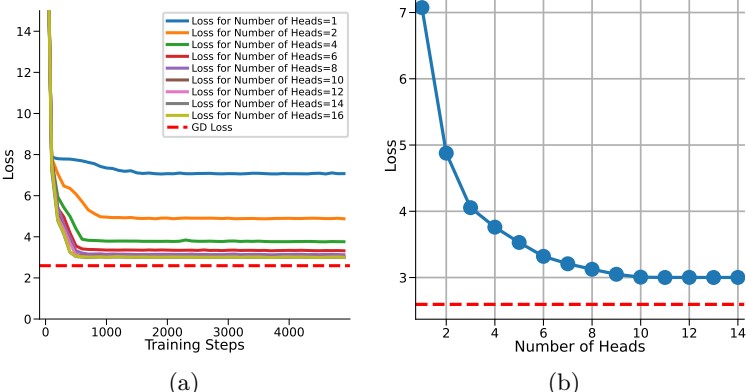

(a)  (b)

Figure 2: **Training loss of multi-head LSA with different numbers of attention heads.** In (a), we visualize the training loss curves for models with different head configurations, each curve shows the expected ICL risk during parameter training (Adam updates of $\{W^Q, W^K, W^V, W^P\}$; no updates at test time). While (b) shows the final trained loss as a function of the number of heads.

Theorem 1. Notably, even at $d + 1$ heads, the multi-head LSA model does not converge to the one-step GD baseline loss, suggesting that while additional heads can capture richer in-context information(Crosbie & Shutova, 2024), they alone are insufficient for achieving full parity with the one-step GD performance in non-zero prior means setting. In other words, one-step GD loss serves as a strict lower bound of the ICL risk for multi-head LSA empirically.

#### 5.1.2 Effect of Prior Mean $\mathbf{w}_\star$ in Multi-Head LSA.

We investigate how the prior mean $\mathbf{w}_\star$, which represents the mean weight of the generated linear function, affects the performance of multi-head LSA when the number of heads is fixed at or above $d + 1$ and $y_q$ is fixed at zero. Fig. 3a shows the loss curves for different values of $\|\mathbf{w}_\star\|$, while Fig. 3b presents the final trained loss as a function of $\|\mathbf{w}_\star\|^2$.

Our results demonstrate that even when the number of heads is sufficiently large (i.e., $\geq d + 1$, reaching the optimal multihead LSA configuration), multi-head LSA only matches the loss of one-step GD when the prior mean $\mathbf{w}_\star$ is zero. For non-zero prior means, a systematic gap remains between Multi-Head LSA and onestep GD. Furthermore, this gap increases linearly with the squared $\ell_2$ norm of the

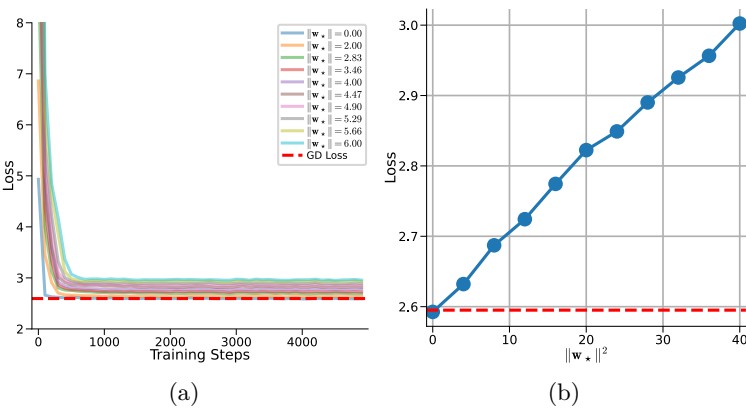

(a)  (b)

Figure 3: **Training loss of multi-head LSA under different prior means $\mathbf{w}_\star$.** (a) Training loss curves for different values of $\|\mathbf{w}_\star\|$. (b) Final trained loss as a function of $\|\mathbf{w}_\star\|^2$. Multi-head LSA matches the one-step GD loss only when $\mathbf{w}_\star = 0$; for $\mathbf{w}_\star \neq 0$ the gap grows approximately linearly with $\|\mathbf{w}_\star\|_2^2$.

prior mean, $\|\mathbf{w}_\star\|^2$, indicating that the prior mean significantly impacts the optimal loss and that larger deviations from zero result in a larger discrepancy from the GD baseline.

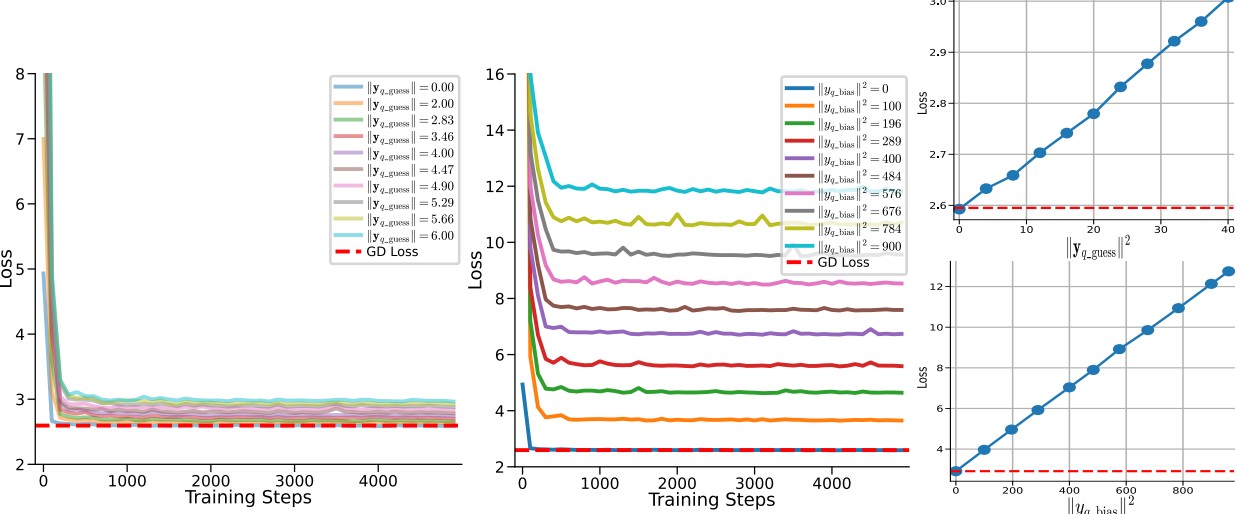

Figure 4: **Training and final loss of multi-head LSA under different initial guess configurations.** **Left** Training loss curves for various $\|y_{\mathrm{q\_bias}}\|^2$, **Middle** Final trained loss as a function of $\|y_{\mathrm{q\_bias}}\|^2$, **Right Upper** Training loss curves for various $\|\mathbf{y}_{\mathrm{q\_guess}}\|$, and **Right Lower** Final trained loss as a function of $\|\mathbf{y}_{\mathrm{q\_guess}}\|^2$. Multi-head LSA reaches the GD loss only when both the linear guess component and the bias vanish ($y_q = \mathbf{w}_\star^\top \mathbf{x}_q$ and no offset).

### 5.1.3 Effect of $y_q$ in LSA

To investigate the effect of the initial guess $y_q$, contained in the embedding matrix equation 2 on the in-context learning ability of multi-head LSA, we decompose $y_q$ into two components:

$$y_q = \mathbf{x}_q^\top \mathbf{y}_{\mathrm{q\_guess}} + y_{\mathrm{q\_bias}}.$$

We set the prior mean $\mathbf{w}_\star$ to zero and number of head is $d + 1$, then conduct two separate experiments: (1) varying $\mathbf{y}_{\mathrm{q\_guess}}$ while fixing $y_{\mathrm{q\_bias}} = 0$, and (2) varying $y_{\mathrm{q\_bias}}$ while fixing $\mathbf{y}_{\mathrm{q\_guess}} = 0$. This allows us to isolate the contribution of each component to the model's behavior.

As shown in Fig. 4, multi-head LSA only converges to the same loss as one-step GD when $\mathbf{y}_{\mathrm{q\_guess}} = 0$ (i.e., equal to the prior mean) and $y_{\mathrm{q\_bias}} = 0$. In all other cases, a systematic gap remains between the loss of multi-head LSA and one-step GD. Moreover, this gap is directly proportional to $\|\mathbf{y}_{\mathrm{q\_guess}}\|^2$ (the squared $\ell_2$-norm of the guessed component) and $\|y_{\mathrm{q\_bias}}\|^2$ (the squared bias term). These findings suggest that deviations in $y_q$ from the

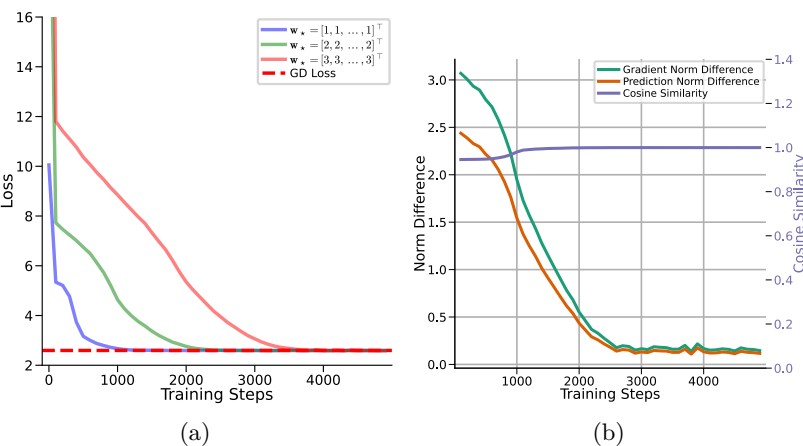

Figure 5: **Training loss and sensitivity analysis of $y_q$-LSA.** (a) Training loss curves of $y_q$-LSA and one-step GD. (b) Model behavior metrics including prediction norm difference, gradient norm difference, and cosine similarity.

optimal initialization introduce a persistent discrepancy in multi-head LSA's performance relative to one-step GD, regardless of the training of multi-head LSA.

### 5.2 $y_q$-LSA

In this section, we aim to empirically validate whether $y_q$-LSA, introduced in Section 4, aligns with one-step GD across different prior settings. Fig. 5 presents the training loss of $y_q$-LSA. Throughout Fig. 5a the dashed "GD Loss" curve is the in-context risk of the predictor obtained by one GD step initialized at the prior mean $\mathbf{w}_0 = \mathbf{w}_\star$: $\mathbf{w}_1 = \mathbf{w}_0 - \frac{\eta}{C}X^\top(X\mathbf{w}_0 - y)$, $\hat{y}_{\text{GD}}(\mathbf{x}_q) = \mathbf{x}_q^\top\mathbf{w}_1$, and the plotted baseline is $\mathcal{R}_{\text{GD-1step}} = \mathbb{E}\left[(\hat{y}_{\text{GD}}(\mathbf{x}_q) - y)^2\right]$.

In Fig. 5a, we compare the convergence of $y_q$-LSA to one-step GD, demonstrating that regardless of the prior configuration, $y_q$-LSA effectively matches the GD solution. Fig. 5b provides a detailed evaluation of prediction norm differences, gradient norm differences (defined in Appendix C.2), and cosine similarity between the models. The results confirm that $y_q$-LSA exhibits strong alignment with one-step GD in both loss convergence and gradient analysis.

### 5.3 LLM experiments

Through theoretical and experimental analysis, we hypothesize that providing an initial guess for the target output during the ICL significantly improves the model's ability to refine its predictions. Specifically, we posit that initial guesses act as a prior for optimization, guiding the model to more accurately. To validate this hypothesis, we conduct experiments leveraging widespread LLMs, demonstrating the efficacy of initial guesses in improving prediction accuracy.

Our experiments utilize Meta-LLaMA-3.1-8B-Instruct (Grattafiori et al., 2024), Qwen/Qwen2.5-7B-Instruct (Yang et al., 2024; Team, 2024) and the STS-Benchmark dataset (English subset) (May, 2021). Each prompt is presented in conjunction with a context comprising 10 labelled examples, where each example included a pair of sentences and its correct similarity score. A lightweight guess model is used to generate initial guesses for both the query and context examples. These guesses are included in the prompts provided to the LLM model, framed as prior guess. The model's task is to predict a similarity score for the query pair, explicitly improving upon the initial guess. For evaluation, we calculate the mean squared error (MSE) between the predicted and true similarity scores, comparing the models with and without initial guesses. More details are in Appendix C.3.

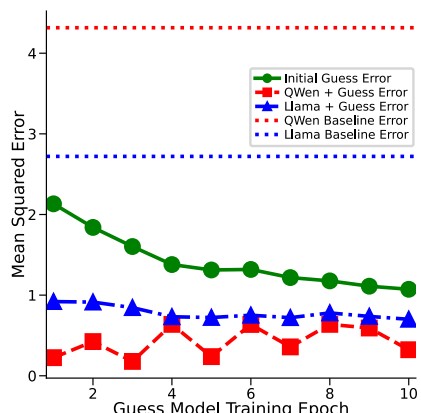

Figure 6: **Error Comparison** Two pre-trained models show consistently improved ICL performance on a sentence similarity task when prompted with a non-trivial initial guess.

The results demonstrate that the inclusion of initial guesses significantly enhances the performance of LLMs in ICL tasks. As shown in Fig. 6, incorporating initial guesses into the context reduce MSE under all experimental conditions. Comparative analysis of the LLaMA and QWen models further underscores the generality of this approach, as both models consistently benefit from the inclusion of initial guesses. These findings follow our hypothesis that initial guesses enhance ICL by providing an initial guess for refinement.

## 6 Conclusion

In this work, we have theoretically and empirically studied the extent to which multi-head LSA approximates GD in ICL, under more realistic assumptions of non-zero prior means. Our analysis establishes that while increasing the number of attention heads to $d + 1$ suffices to reach the minimal ICL risk in the linear setting, the model fundamentally fails to reach a stationary point when the prior mean is non-zero and context size

grows. This limitation is further connected with the initial guess $y_q$, whose misalignment with the prior induces a persistent optimality gap, even when the number of heads is sufficient. To solve this, we introduce $y_q$-LSA, an LSA variant with a trainable initial guess, and show both theoretically and empirically that it bridges the gap between LSA and one-step GD in linear regression. Finally, we illustrate that incorporating an initial guess also benefits ICL in large language models, showing how this approach can be also used in more common settings.

**Limitations.** While our analysis is limited to linear regression tasks and simplified architectures without nonlinearities, normalization, or softmax, these assumptions are standard across much of the theoretical literature on in-context learning and mechanistic interpretation of transformers. The theoretical results rely on the infinite-context limit, which, although analytically tractable, diverges from practical settings where context size is finite. Additionally, while $y_q$-LSA closes the gap with one-step GD in controlled experiments, its applicability to complex real-world tasks remains contingent on effective mechanisms for estimating or learning initial guesses. The LLM experiments suggest empirical benefits, but further exploration is required to assess generalizability across diverse tasks, model families, and training regimes.

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

# A   Proofs of Section 3

For the sake of completeness and self-containment, we restate the theorems and lemmas shown in Section 3 and provide their full proof in this section.

## A.1 Proof of Theorem 1

First, let's redefine the notations used in Theorem 1 and restate the theorem. We write the input of a model as an *embedding matrix* given by

$$\boldsymbol{E} \stackrel{\text{def}}{=} \begin{bmatrix} \boldsymbol{X}^{\top} & \mathbf{x}_q \\ \mathbf{y}^{\top} & y_q \end{bmatrix} \in \mathbb{R}^{(d+1)\times(C+1)}, \tag{13}$$

where $\boldsymbol{X}, \mathbf{y}, \mathbf{x}_q, y_q$ are defined in Section 2. The multi-head linear-self attention (LSA) function is defined as

$$f_{\mathsf{H-LSA}}(\boldsymbol{E}) \stackrel{\text{def}}{=} \left[ \boldsymbol{E} + \sum_{h=1}^{H} \text{head}_h(\boldsymbol{E}) \right]_{-1,-1}, \tag{14}$$

where the output of each transformer head is defined as

$$\text{head}_h(\boldsymbol{E}) \stackrel{\text{def}}{=} \tfrac{1}{C} \mathbf{W}_h^P \mathbf{W}_h^V \boldsymbol{E} \mathbf{W}^M \left( \boldsymbol{E}^{\top} (\mathbf{W}_h^K)^{\top} \mathbf{W}_h^Q \boldsymbol{E} \right), \quad h \in [H]. \tag{15}$$

The trainable parameters $\mathbf{W}_h^K, \mathbf{W}_h^Q, \mathbf{W}_h^P$ and $\mathbf{W}_h^V$ are specific to the $h$-th head, and $\mathbf{W}^M \stackrel{\text{def}}{=} \begin{bmatrix} \boldsymbol{I}_C & 0 \\ 0 & 0 \end{bmatrix}$ is a mask matrix, to ignore the query token when computing the attention scores. Let's define by

$$\mathcal{F}_{H-\mathsf{LSA}} \stackrel{\text{def}}{=} \left\{ f_{\mathsf{H-LSA}} \;\middle|\; \left\{ \mathbf{W}_h^K, \mathbf{W}_h^Q, \mathbf{W}_h^V, \mathbf{W}_h^P \right\}_{h=1}^{H} \right\}$$

the hypothesis class associated with multi-head LSA models with $H$ heads. Finally, we measure the ICL risk of a model $f$ by the mean squared error,

$$\mathcal{R}(f) \stackrel{\text{def}}{=} \mathbb{E}[(f(\boldsymbol{E}) - y)^2], \tag{16}$$

where the expectation is taken over the data distribution (and effectively over the embedding matrix $\boldsymbol{E}$ defined in equation 13).

Now we are ready to restate and prove Theorem 1.

**Theorem 1.** *Let $d \in \mathbb{N}$, and consider the hypothesis classes $\mathcal{F}_{(d+1)-\mathsf{LSA}}$ and $\mathcal{F}_{(d+2)-\mathsf{LSA}}$ corresponding to multi-head LSA models with $H = d+1$ and $H = d+2$ attention heads, respectively. Then*

$$\inf_{f \in \mathcal{F}_{(d+1)-\mathsf{LSA}}} \mathcal{R}(f) = \inf_{f \in \mathcal{F}_{(d+2)-\mathsf{LSA}}} \mathcal{R}(f),$$

*where $\mathcal{R}(f)$ is the ICL risk defined in Eq. (4).*

*Proof.* To simplify the notation, let's introduce a couple of additional definitions. For each head $h \in [H]$, the product of the output projection $\mathbf{W}_h^P$ and the value projection $\mathbf{W}_h^V$ can be written without loss of generality as

$$\mathbf{W}_h^P \mathbf{W}_h^V \stackrel{\text{def}}{=} \begin{bmatrix} * \\ \mathbf{b}_h^{\top} \end{bmatrix} \in \mathbb{R}^{(d+1)\times(d+1)},$$

where $\mathbf{b}_h \in \mathbb{R}^{d+1}$ is the last row of the matrix, and the block $*$ denotes entries that have no influence on the ICL risk. Then, let's rewrite the product of the key and query matrices as

$$(\mathbf{W}_h^K)^{\top} \mathbf{W}_h^Q \stackrel{\text{def}}{=} \boldsymbol{A}_h \in \mathbb{R}^{(d+1)\times(d+1)},$$

and denote its column decomposition by

$$\boldsymbol{A}_h = \begin{bmatrix} \mathbf{a}_1^h & \cdots & \mathbf{a}_{d+1}^h \end{bmatrix},$$

where $\mathbf{a}_i^h \in \mathbb{R}^{d+1}$ for each $i \in [d+1]$.

With this notation, the contribution of all heads to the attention mechanism can be expressed in terms of the matrices

$$\boldsymbol{M}_i \stackrel{\text{def}}{=} \sum_{h=1}^{H} \mathbf{b}_h (\mathbf{a}_i^h)^\top \in \mathbb{R}^{(d+1) \times (d+1)}, \qquad i \in [d+1].$$

Each $\boldsymbol{M}_i$ is a $(d+1) \times (d+1)$ real matrix. The space of such matrices, $\mathbb{R}^{(d+1) \times (d+1)}$, has dimension $(d+1)^2$.

The collection

$$(\boldsymbol{M}_1, \boldsymbol{M}_2, \ldots, \boldsymbol{M}_{d+1})$$

is thus an element of the Cartesian product

$$\left(\mathbb{R}^{(d+1) \times (d+1)}\right)^{d+1}.$$

with dimension $\dim\left(\left(\mathbb{R}^{(d+1) \times (d+1)}\right)^{d+1}\right) = (d+1)^3$. Hence, the set of all possible tuples $(\boldsymbol{M}_1, \ldots, \boldsymbol{M}_{d+1})$ can be identified with a vector space of dimension $(d+1)^3$.

We now compute the number of parameters available per head. For a fixed head $h$, the parameters that influence the construction of $\boldsymbol{M}_i$ are (1) the vector $\mathbf{b}_h$ which contributes $(d+1)$ free parameters, (2) the family of vectors $\mathbf{a}_1^h, \ldots, \mathbf{a}_{d+1}^h$, which contributes $(d+1)(d+1)$ free parameters. Therefore, in total one head contributes $(d+1) + (d+1)(d+1) = (d+1)(d+2)$ degrees of freedom. With $H$ heads in total, the dimension of the parameter space $\Omega_H$ is $\dim(\Omega_H) = H(d+1)(d+2)$.

Suppose $H \geq d+1$. Then

$$H(d+1)(d+2) \geq (d+1)(d+1)(d+2).$$

Since $(d+2) \geq (d+1)$, we obtain

$$H(d+1)(d+2) \geq (d+1)^3.$$

This inequality shows that, when $H \geq d+1$, the parameter space has dimension at least as large as the target space. In particular, there is no dimensional obstruction to surjectivity of the mapping from parameters $(\mathbf{b}_h, \mathbf{a}_i^h)$ to matrices $(\boldsymbol{M}_1, \ldots, \boldsymbol{M}_{d+1})$.

To demonstrate that the mapping is indeed surjective once $H \geq d+1$, we now construct explicitly any desired collection of matrices $(\boldsymbol{M}_1, \ldots, \boldsymbol{M}_{d+1})$.

Fix $i \in [d+1]$. Let $\mathbf{e}_1, \ldots, \mathbf{e}_{d+1}$ denote the standard basis vectors of $\mathbb{R}^{d+1}$. For each $h \in [d+1]$, set

$$\mathbf{b}_h = \mathbf{e}_h, \qquad \mathbf{a}_i^h = \boldsymbol{M}_i[h],$$

where $\boldsymbol{M}_i[h]$ denotes the $h$-th row of the matrix $\boldsymbol{M}_i$. For $h > d+1$, we may set $\mathbf{b}_h = 0$ and $\mathbf{a}_i^h = 0$, so that those heads contribute nothing. With this choice of parameters,

$$\sum_{h=1}^{d+1} \mathbf{b}_h (\mathbf{a}_i^h)^\top = \sum_{h=1}^{d+1} \mathbf{e}_h \left(\boldsymbol{M}_i[h]\right)^\top = \boldsymbol{M}_i.$$

Thus, every $\boldsymbol{M}_i$ is exactly reproduced, and therefore every tuple $(\boldsymbol{M}_1, \ldots, \boldsymbol{M}_{d+1})$ is realizable when $H \geq d+1$.

We have shown that with $H = d+1$ heads, the model can realize any element of the target space, and therefore the hypothesis class is saturated. Adding additional heads $H > d+1$ cannot enlarge the class of realizable functions. For this reason, for any $H \geq d+1$, we have

$$\inf_{f \in \mathcal{F}_{(d+2)-\mathsf{LSA}}} \mathcal{R}(f) \leq \inf_{f \in \mathcal{F}_{(d+1)-\mathsf{LSA}}} \mathcal{R}(f).$$

Finally, observe that $\mathcal{F}_{(d+1)-\mathsf{LSA}} \subseteq \mathcal{F}_{(d+2)-\mathsf{LSA}}$, since a $(d+1)$-head model can be viewed as a $(d+2)$-head model with the additional head parameters set to zero. Consequently, it follows that the only possibility is that

$$\inf_{f \in \mathcal{F}_{(d+1)-\mathsf{LSA}}} \mathcal{R}(f) = \inf_{f \in \mathcal{F}_{(d+2)-\mathsf{LSA}}} \mathcal{R}(f),$$

which concludes the proof.

$\square$

## A.2   Proof of Theorem 2

**Theorem 2.** *Let $H \in \mathbb{N}$ and consider the hypothesis class $\mathcal{F}_{H-\mathsf{LSA}}$ of multi-head LSA models with context size $C \to \infty$. Then the in-context learning risk $\mathcal{R}(f)$ admits no non-trivial stationary point in parameter space. More precisely,*

$$\nabla \mathcal{R}(f) \neq 0 \quad \text{for all } f \in \mathcal{F}_{H-\mathsf{LSA}}$$

*for every choice of parameters $\{\mathbf{W}_h^K, \mathbf{W}_h^Q, \mathbf{W}_h^V, \mathbf{W}_h^P\}_{h=1}^H$, except in the case where the prior mean vector vanishes, $\mathbf{w}_\star = 0$.*

The proof of Theorem 2 is based on the analysis of Ahn et al. (2023).

*Proof.* **Step 1: Simplify the risk function and compute its gradient**

We first derive explicitly the expression of multi-head LSA's ICL risk and simplify it. The key idea is to decompose the ICL risk into components. That is,

$$
\begin{aligned}
\mathcal{R}(f_{H-\mathsf{LSA}}) &\overset{equation\ 4}{=} \mathbb{E}\left[(f_{H-\mathsf{LSA}}(\boldsymbol{E}) - y)^2\right] \quad \text{with } y = \widehat{\mathbf{w}}^\top \mathbf{x}_q \text{ and } \widehat{\mathbf{w}} \sim \mathcal{N}(\mathbf{w}_\star, \boldsymbol{I}_d), \\
&\overset{equation\ 6}{=} \mathbb{E}\left[\left(\left[\boldsymbol{E} + \sum_{h=1}^H \text{head}_h(\boldsymbol{E})\right]_{-1,-1} - \widehat{\mathbf{w}}^\top \mathbf{x}_q\right)^2\right] \\
&\overset{equation\ 5}{=} \mathbb{E}\left[\left(\left[\boldsymbol{E} + \frac{1}{C}\sum_{h=1}^H \mathbf{W}_h^P \mathbf{W}_h^V \boldsymbol{E} \mathbf{W}^M \left(\boldsymbol{E}^\top (\mathbf{W}_h^K)^\top \mathbf{W}_h^Q \boldsymbol{E}\right)\right]_{-1,-1} - \widehat{\mathbf{w}}^\top \mathbf{x}_q\right)^2\right].
\end{aligned}
$$

Since the prediction of $f_{H-\mathsf{LSA}}$ is the bottom right entry of the output matrix, only the last row of the product $\mathbf{W}_h^P \mathbf{W}_h^V$ contributes to the prediction. Therefore, we write

$$\mathbf{W}_h^P \mathbf{W}_h^V \overset{\text{def}}{=} \begin{bmatrix} * \\ \mathbf{b}_h^\top \end{bmatrix} \in \mathbb{R}^{(d+1)\times(d+1)},$$

where $\mathbf{b}_h \in \mathbb{R}^{d+1}$ for all $h \in [H]$, and $*$ denotes entries that do not affect the ICL risk.

To simplify the computation, we also rewrite the product $(\mathbf{W}_h^K)^\top \mathbf{W}_h^Q$ and the embedding matrix $\boldsymbol{E}$ as

$$
\begin{aligned}
(\mathbf{W}_h^K)^\top \mathbf{W}_h^Q &\overset{\text{def}}{=} \boldsymbol{A}_h \in \mathbb{R}^{(d+1)\times(d+1)}, \\
\boldsymbol{E} &\overset{\text{def}}{=} \begin{bmatrix} \mathbf{z}_1 & \mathbf{z}_2 & \cdots & \mathbf{z}_C & \mathbf{z}_{C+1} \end{bmatrix} \in \mathbb{R}^{(d+1)\times(C+1)},
\end{aligned}
$$

where

$$
\begin{aligned}
\boldsymbol{A}_h &\overset{\text{def}}{=} \begin{bmatrix} \mathbf{a}_1^h & \mathbf{a}_2^h & \cdots & \mathbf{a}_{d+1}^h \end{bmatrix} \quad \text{with } \mathbf{a}_1^h, \cdots, \mathbf{a}_{d+1}^h \in \mathbb{R}^{d+1}, \\
\mathbf{z}_i &\overset{\text{def}}{=} \begin{bmatrix} \mathbf{x}_i \\ y_i \end{bmatrix} \in \mathbb{R}^{d+1} \quad \text{for all } i \in [C], \quad \text{and} \quad \mathbf{z}_{C+1} \overset{\text{def}}{=} \begin{bmatrix} \mathbf{x}_q \\ y_q \end{bmatrix} \in \mathbb{R}^{d+1}.
\end{aligned}
$$

We define

$$\boldsymbol{G} \overset{\text{def}}{=} \frac{1}{C}\sum_{i=1}^C \mathbf{z}_i \mathbf{z}_i^\top = \frac{1}{C} \boldsymbol{E} \mathbf{W}^M \boldsymbol{E}^\top \in \mathbb{R}^{(d+1)\times(d+1)} \quad \text{and} \quad \widehat{\mathbf{w}} \overset{\text{def}}{=} \mathbf{w}_\star + \boldsymbol{\epsilon},$$

where $\boldsymbol{\epsilon} \in \mathbb{R}^d \sim \mathcal{N}(0, \boldsymbol{I}_d)$ is the noise.

Then the ICL risk can be written as

$$
\mathcal{R}(f_{\mathsf{H-LSA}}) = \mathbb{E}\left[\left(y_q + \sum_{h=1}^{H}\mathbf{b}_h^\top \boldsymbol{G}\boldsymbol{A}_h\mathbf{z}_{C+1} - \widehat{\mathbf{w}}^\top\mathbf{x}_q\right)^2\right]
$$

$$
= \mathbb{E}\left[\left(y_q + \sum_{h=1}^{H}\mathbf{b}_h^\top \boldsymbol{G}\begin{bmatrix}\mathbf{a}_1^h & \mathbf{a}_2^h & \cdots & \mathbf{a}_{d+1}^h\end{bmatrix}\begin{bmatrix}\mathbf{x}_q \\ y_q\end{bmatrix} - \widehat{\mathbf{w}}^\top\mathbf{x}_q\right)^2\right]
$$

$$
= \mathbb{E}\left[\left(y_q + \sum_{h=1}^{H}\left(\sum_{i=1}^{d}\mathbf{b}_h^\top \boldsymbol{G}\mathbf{a}_i^h\mathbf{x}_q[i]\right) + \mathbf{b}_h^\top \boldsymbol{G}\mathbf{a}_{d+1}^h y_q - \widehat{\mathbf{w}}^\top\mathbf{x}_q\right)^2\right],
$$

where $\mathbf{x}_q[i]$ is the $i$-th coordinate of the vector $\mathbf{x}_q$.

Furthermore, we know that, for all $h \in [H]$ and $i \in [d+1]$,

$$
\mathbf{b}_h^\top \boldsymbol{G}\mathbf{a}_i^h \in \mathbb{R} = \mathrm{Tr}\left(\mathbf{b}_h^\top \boldsymbol{G}\mathbf{a}_i^h\right) = \mathrm{Tr}\left(\boldsymbol{G}\mathbf{a}_i^h\mathbf{b}_h^\top\right) = \left\langle \boldsymbol{G}, \mathbf{b}_h(\mathbf{a}_i^h)^\top\right\rangle,
$$

where $\langle \boldsymbol{U}, \boldsymbol{V}\rangle \overset{\text{def}}{=} \mathrm{Tr}\left(\boldsymbol{U}\boldsymbol{V}^\top\right)$ is the Frobenius inner product for any squared matrices $\boldsymbol{U}$ and $\boldsymbol{V}$.

Hence, by using the linearity of the Frobenius inner product, we rewrite the ICL risk as

$$
\mathcal{R}(f_{\mathsf{H-LSA}})
$$

$$
= \mathbb{E}\left[\left(y_q + \sum_{h=1}^{H}\left\langle \boldsymbol{G}, \mathbf{b}_h(\mathbf{a}_{d+1}^h)^\top\right\rangle y_q + \sum_{i=1}^{d}\sum_{h=1}^{H}\left(\left\langle \boldsymbol{G}, \mathbf{b}_h(\mathbf{a}_i^h)^\top\right\rangle - \widehat{\mathbf{w}}[i]\right)\mathbf{x}_q[i]\right)^2\right]
$$

$$
= \mathbb{E}\left[\left(\left(1 + \left\langle \boldsymbol{G}, \sum_{h=1}^{H}\mathbf{b}_h(\mathbf{a}_{d+1}^h)^\top\right\rangle\right)y_q + \sum_{i=1}^{d}\left(\left\langle \boldsymbol{G}, \sum_{h=1}^{H}\mathbf{b}_h(\mathbf{a}_i^h)^\top\right\rangle - \widehat{\mathbf{w}}[i]\right)\mathbf{x}_q[i]\right)^2\right],
$$

where $\widehat{\mathbf{w}}[i]$ is the $i$-th coordinate of the vector $\widehat{\mathbf{w}}$.

By reparametrizing the ICL risk, using a composite function, we have

$$
\mathcal{R}(f_{\mathsf{H-LSA}}) = \mathbb{E}_{\boldsymbol{G},\widehat{\mathbf{w}},\mathbf{x}_q}\left[\left((1 + \langle \boldsymbol{G}, \boldsymbol{M}_{d+1}\rangle)y_q + \sum_{i=1}^{d}\left(\langle \boldsymbol{G}, \boldsymbol{M}_i\rangle - \widehat{\mathbf{w}}[i]\right)\mathbf{x}_q[i]\right)^2\right], \tag{17}
$$

where

$$
\boldsymbol{M}_i \overset{\text{def}}{=} \sum_{h=1}^{H}\mathbf{b}_h(\mathbf{a}_i^h)^\top \in \mathbb{R}^{(d+1)\times(d+1)}, \qquad \text{for all } i \in [d+1].
$$

Recall $\mathbf{x}_q \sim \mathcal{N}(0, \boldsymbol{I}_d)$. Thus, both $\boldsymbol{G}$ and $\widehat{\mathbf{w}}$ are independent to $\mathbf{x}_q[i]$ for all $i \in [d]$, and $\mathbf{x}_q[i] \sim \mathcal{N}(0,1)$ are i.i.d. Expanding equation 17 yields

$$
\mathcal{R}(f_{\mathsf{H-LSA}}) = \mathbb{E}_{\boldsymbol{G}}\left[(1 + \langle \boldsymbol{G}, \boldsymbol{M}_{d+1}\rangle)^2 y_q^2\right] + \sum_{i=1}^{d}\mathbb{E}_{\boldsymbol{G},\widehat{\mathbf{w}}}\left[(\langle \boldsymbol{G}, \boldsymbol{M}_i\rangle - \widehat{\mathbf{w}}[i])^2\right]\mathbb{E}_{\mathbf{x}_q}\left[\mathbf{x}_q[i]^2\right]
$$

$$
= \mathbb{E}_{\boldsymbol{G}}\left[(1 + \langle \boldsymbol{G}, \boldsymbol{M}_{d+1}\rangle)^2 y_q^2\right] + \sum_{i=1}^{d}\mathbb{E}_{\boldsymbol{G},\widehat{\mathbf{w}}}\left[(\langle \boldsymbol{G}, \boldsymbol{M}_i\rangle - \widehat{\mathbf{w}}[i])^2\right]
$$

$$
= \sum_{i=1}^{d+1}\mathcal{L}_i(\boldsymbol{M}_i), \tag{18}
$$

where

$$\mathcal{L}_i(\boldsymbol{M}_i) \stackrel{\text{def}}{=} \mathbb{E}_{\boldsymbol{G},\widehat{\mathbf{w}}}\left[(\langle \boldsymbol{G}, \boldsymbol{M}_i\rangle - \widehat{\mathbf{w}}[i])^2\right] \quad \text{for all } i \in [d],$$

$$\mathcal{L}_{d+1}(\boldsymbol{M}_{d+1}) \stackrel{\text{def}}{=} \mathbb{E}_{\boldsymbol{G}}\left[(1 + \langle \boldsymbol{G}, \boldsymbol{M}_{d+1}\rangle)^2 y_q^2\right].$$

Thus, the ICL risk equation 18 is decomposed into $(d+1)$ separated components $\mathcal{L}_i$ with $i \in [d+1]$. Each component is a function of $\boldsymbol{M}_i$. To compute the gradient of $\mathcal{R}(f_{\mathsf{H-LSA}})$, we can first compute the gradient of each component with respect to $\boldsymbol{M}_i$ for $i \in [d]$. That is,

$$\nabla_{\boldsymbol{M}_i}\mathcal{L}_i(\boldsymbol{M}_i) = 2\mathbb{E}_{\boldsymbol{G},\widehat{\mathbf{w}}}\left[\langle \boldsymbol{G}, \boldsymbol{M}_i\rangle \boldsymbol{G}\right] - 2\mathbb{E}_{\boldsymbol{G},\widehat{\mathbf{w}}}\left[\widehat{\mathbf{w}}[i]\boldsymbol{G}\right], \qquad \text{for } i \in [d], \tag{19}$$

$$\nabla_{\boldsymbol{M}_{d+1}}\mathcal{L}_{d+1}(\boldsymbol{M}_{d+1}) = 2y_q^2\mathbb{E}_{\boldsymbol{G}}\left[(1 + \langle \boldsymbol{G}, \boldsymbol{M}_{d+1}\rangle)\boldsymbol{G}\right]. \tag{20}$$

**Step 2: Compute $\mathbb{E}\left[\langle \boldsymbol{G}, \boldsymbol{M}_i\rangle \boldsymbol{G}\right]$, $\mathbb{E}\left[\widehat{\mathbf{w}}[i]\boldsymbol{G}\right]$ in equation 19**

Recall that $\widehat{\mathbf{w}} \sim \mathcal{N}(\mathbf{w}_\star, \boldsymbol{I}_d)$ and $\mathbf{x}_j \overset{\text{i.i.d.}}{\sim} \mathcal{N}(0, \boldsymbol{I}_d)$ are independent for all $j \in [C]$, $y_j = \widehat{\mathbf{w}}^\top \mathbf{x}_j$, and $\widehat{\mathbf{w}} = \mathbf{w}_\star + \boldsymbol{\epsilon}$ with $\boldsymbol{\epsilon} \sim \mathcal{N}(0, \boldsymbol{I}_d)$.

For $\mathbb{E}_{\boldsymbol{G},\widehat{\mathbf{w}}}\left[\widehat{\mathbf{w}}[i]\boldsymbol{G}\right]$ in equation 19 with $i \in [d]$, we have

$$\mathbb{E}_{\boldsymbol{G},\widehat{\mathbf{w}}}\left[\widehat{\mathbf{w}}[i]\boldsymbol{G}\right] = \frac{1}{C}\sum_{j=1}^{C}\begin{bmatrix} \mathbb{E}_{\widehat{\mathbf{w}},\mathbf{x}_j}[\widehat{\mathbf{w}}[i] \cdot \mathbf{x}_j\mathbf{x}_j^\top] & \mathbb{E}_{\widehat{\mathbf{w}},\mathbf{x}_j}[\widehat{\mathbf{w}}[i] \cdot y_j\mathbf{x}_j] \\ \mathbb{E}_{\widehat{\mathbf{w}},\mathbf{x}_j}[\widehat{\mathbf{w}}[i] \cdot y_j\mathbf{x}_j^\top] & \mathbb{E}_{\widehat{\mathbf{w}},\mathbf{x}_j}[\widehat{\mathbf{w}}[i] \cdot y_j^2] \end{bmatrix}.$$

In particular, for each block of the above matrix, we have

$$\mathbb{E}_{\widehat{\mathbf{w}},\mathbf{x}_j}[\widehat{\mathbf{w}}[i] \cdot \mathbf{x}_j\mathbf{x}_j^\top] = \mathbb{E}_{\widehat{\mathbf{w}}}[\widehat{\mathbf{w}}[i]]\,\mathbb{E}_{\mathbf{x}_j}\left[\mathbf{x}_j\mathbf{x}_j^\top\right] = \mathbf{w}_\star[i]\boldsymbol{I}_d,$$

$$\mathbb{E}_{\widehat{\mathbf{w}},\mathbf{x}_j}[\widehat{\mathbf{w}}[i] \cdot y_j\mathbf{x}_j] = \mathbb{E}_{\widehat{\mathbf{w}},\mathbf{x}_j}\left[\widehat{\mathbf{w}}[i]\widehat{\mathbf{w}}^\top\mathbf{x}_j\mathbf{x}_j\right]$$

$$= \mathbb{E}_{\boldsymbol{\epsilon},\mathbf{x}_j}\left[(\mathbf{w}_\star[i] + \boldsymbol{\epsilon}[i])(\mathbf{w}_\star + \boldsymbol{\epsilon})^\top\mathbf{x}_j\mathbf{x}_j\right] = \mathbf{w}_\star[i]\mathbf{w}_\star + \mathbf{e}_i,$$

$$\mathbb{E}_{\widehat{\mathbf{w}},\mathbf{x}_j}[\widehat{\mathbf{w}}[i] \cdot y_j^2] = \mathbb{E}_{\widehat{\mathbf{w}},\mathbf{x}_j}[\widehat{\mathbf{w}}[i]\widehat{\mathbf{w}}^\top\mathbf{x}_j\mathbf{x}_j^\top\widehat{\mathbf{w}}] = \mathbb{E}_{\widehat{\mathbf{w}}}\left[\widehat{\mathbf{w}}[i]\widehat{\mathbf{w}}^\top\widehat{\mathbf{w}}\right]$$

$$= \mathbb{E}_{\boldsymbol{\epsilon}}\left[(\mathbf{w}_\star[i] + \boldsymbol{\epsilon}[i])(\mathbf{w}_\star + \boldsymbol{\epsilon})^\top(\mathbf{w}_\star + \boldsymbol{\epsilon})\right] = \mathbf{w}_\star[i](\|\mathbf{w}_\star\|^2 + d + 2),$$

where $\mathbf{e}_i$ denotes the standard basis vector with zeros in all coordinates except the $i$-th position, where the value is 1.

Combining the above three components, we have

$$\mathbb{E}_{\boldsymbol{G},\widehat{\mathbf{w}}}\left[\widehat{\mathbf{w}}[i]\boldsymbol{G}\right] = \begin{bmatrix} \mathbf{w}_\star[i]\boldsymbol{I}_d & \mathbf{w}_\star[i]\mathbf{w}_\star + \mathbf{e}_i \\ (\mathbf{w}_\star[i]\mathbf{w}_\star + \mathbf{e}_i)^\top & \mathbf{w}_\star[i](\|\mathbf{w}_\star\|^2 + d + 2) \end{bmatrix}. \tag{21}$$

Now we compute $\mathbb{E}\left[\langle \boldsymbol{G}, \boldsymbol{M}_i\rangle \boldsymbol{G}\right]$ for $i \in [d]$.

We start by calculating the expected value of the product of elements in matrix $\boldsymbol{G}$. That is, for all $m, n, p, q \in [d+1]$,

$$\mathbb{E}\left[\boldsymbol{G}_{mn}\boldsymbol{G}_{pq}\right] = \frac{1}{C^2}\sum_{j=1}^{C}\sum_{k=1}^{C}\mathbb{E}\left[\mathbf{z}_j[m]\mathbf{z}_j[n]\mathbf{z}_k[p]\mathbf{z}_k[q]\right],$$

where $\boldsymbol{G}_{mn}$ is the value of matrix $\boldsymbol{G}$ in $m$-th row and $n$-th column position for all $m, n \in [d+1]$. By expanding the summation, we have

$$\mathbb{E}\left[\boldsymbol{G}_{mn}\boldsymbol{G}_{pq}\right] = \frac{1}{C^2}\sum_{\substack{1 \le j,k \le C \\ j \ne k}}\mathbb{E}\left[\mathbf{z}_j[m]\mathbf{z}_j[n]\mathbf{z}_k[p]\mathbf{z}_k[q]\right] + \frac{1}{C}\mathbb{E}\left[\mathbf{z}_1[m]\mathbf{z}_1[n]\mathbf{z}_1[p]\mathbf{z}_1[q]\right]$$

$$= \frac{C(C-1)}{C^2}\mathbb{E}\left[\mathbf{z}_1[m]\mathbf{z}_1[n]\right]\mathbb{E}\left[\mathbf{z}_2[p]\mathbf{z}_2[q]\right] + \frac{1}{C}\mathbb{E}\left[\mathbf{z}_1[m]\mathbf{z}_1[n]\mathbf{z}_1[p]\mathbf{z}_1[q]\right]$$

$$\approx \mathbb{E}\left[\mathbf{z}_1[m]\mathbf{z}_1[n]\right]\mathbb{E}\left[\mathbf{z}_2[p]\mathbf{z}_2[q]\right], \qquad \text{when } C \longrightarrow \infty.$$

To compute $\mathbb{E}\left[\mathbf{z}_1[m]\mathbf{z}_1[n]\right]$,

1. For $m, n \in [d]$, we have $\mathbf{z}_1[m] = \mathbf{x}_1[n]$ and $\mathbf{z}_1[n] = \mathbf{x}_1[n]$. Thus, $\mathbb{E}\left[\mathbf{z}_1[m]\mathbf{z}_1[n]\right] = \delta_{mn}$, where $\delta$ is the Kronecker delta.

2. For $m \in [d]$ and $n = d + 1$, we have $\mathbf{z}_1[n] = y_1$. Thus, $\mathbb{E}\left[\mathbf{z}_1[m]\mathbf{z}_1[n]\right] = \mathbb{E}\left[\mathbf{x}_1[m]\mathbf{x}_1^\top \widehat{\mathbf{w}}\right] = \mathbf{w}_\star[m]$.

3. For $m = n = d + 1$, we have $\mathbb{E}\left[\mathbf{z}_1[m]\mathbf{z}_1[n]\right] = \mathbb{E}\left[\widehat{\mathbf{w}}^\top \mathbf{x}_1 \mathbf{x}_1^\top \widehat{\mathbf{w}}\right] = \mathbb{E}\left[\widehat{\mathbf{w}}^\top \widehat{\mathbf{w}}\right] = \|\mathbf{w}_\star\|^2 + d$.

We denote

$$\boldsymbol{M} \overset{\text{def}}{=} \begin{bmatrix} \boldsymbol{I}_d & \mathbf{w}_\star \\ \mathbf{w}_\star^\top & \|\mathbf{w}_\star\|^2 + d \end{bmatrix} \in \mathbb{R}^{(d+1)\times(d+1)}. \tag{22}$$

By using equation 22, when $C \longrightarrow \infty$, we have

$$\mathbb{E}\left[\boldsymbol{G}_{mn}\boldsymbol{G}_{pq}\right] = \boldsymbol{M}_{mn}\boldsymbol{M}_{pq}. \tag{23}$$

By linearity of the Frobenius inner product, we have

$$\mathbb{E}\left[\langle \boldsymbol{G}, \boldsymbol{M}_i\rangle \boldsymbol{G}\right] = \langle \boldsymbol{M}, \boldsymbol{M}_i\rangle \boldsymbol{M}. \tag{24}$$

Combining the above equation with equation 21, equation 19 becomes

$$\begin{aligned} \nabla_{\boldsymbol{M}_i} \mathcal{L}_i(\boldsymbol{M}_i) &= 2\langle \boldsymbol{M}, \boldsymbol{M}_i\rangle \boldsymbol{M} - 2\begin{bmatrix} \mathbf{w}_\star[i]\boldsymbol{I}_d & \mathbf{w}_\star[i]\mathbf{w}_\star + \mathbf{e}_i \\ (\mathbf{w}_\star[i]\mathbf{w}_\star + \mathbf{e}_i)^\top & \mathbf{w}_\star[i](\|\mathbf{w}_\star\|^2 + d + 2) \end{bmatrix} \\ &= 2\langle \boldsymbol{M}, \boldsymbol{M}_i\rangle \boldsymbol{M} - 2\mathbf{w}_\star[i]\boldsymbol{M} - 2\boldsymbol{N} \\ &= (2\langle \boldsymbol{M}, \boldsymbol{M}_i\rangle - 2\mathbf{w}_\star[i])\boldsymbol{M} - 2\boldsymbol{N}, \end{aligned} \tag{25}$$

where

$$\boldsymbol{N} \overset{\text{def}}{=} \begin{bmatrix} 0 & \mathbf{e}_i \\ \mathbf{e}_i^\top & 2\mathbf{w}_\star[i] \end{bmatrix}.$$

Notice that $\boldsymbol{M}$ is full rank and the rank of $\boldsymbol{N}$ is smaller or equal to 2. Thus, for any $\boldsymbol{M}_i \in \mathbb{R}^{(d+1)\times(d+1)}$, we have

$$\nabla_{\boldsymbol{M}_i} \mathcal{L}_i(\boldsymbol{M}_i) \neq 0.$$

$\square$

### A.3 Finite-context corrections and dependence on the context size $C$

In the proof of Theorem 2 we work in the infinite-context limit $C \to \infty$ and use

$$\mathbb{E}[\mathbf{G_{mn}G_{pq}}] = \mathbf{M_{mn}M_{pq}}, \tag{26}$$

where $\mathbf{G} \in \mathbb{R}^{(\mathbf{d+1})\times(\mathbf{d+1})}$ defined in equations (22)–(24). For completeness, we now make explicit how equation 26 is obtained from the finite-$C$ expression and how the resulting gradients are modified when $C$ is finite.

Recall that

$$\mathbf{G} \overset{\text{def}}{=} \frac{1}{\mathbf{C}} \sum_{\mathbf{j=1}}^{\mathbf{C}} \mathbf{z_j z_j^\top}, \qquad \mathbf{z_j} \overset{\text{def}}{=} \begin{bmatrix} \mathbf{x_j} \\ y_j \end{bmatrix} \in \mathbb{R}^{\mathbf{d+1}}.$$

For any indices $m, n, p, q \in [d + 1]$ we have

$$\mathbf{G_{mn}} = \frac{1}{\mathbf{C}} \sum_{\mathbf{j=1}}^{\mathbf{C}} \mathbf{z_j[m]z_j[n]}, \qquad \mathbf{G_{pq}} = \frac{1}{\mathbf{C}} \sum_{\mathbf{k=1}}^{\mathbf{C}} \mathbf{z_k[p]z_k[q]},$$

and therefore

$$\mathbf{G_{mn}G_{pq}} = \frac{1}{\mathbf{C^2}} \sum_{\mathbf{j=1}}^{\mathbf{C}} \sum_{\mathbf{k=1}}^{\mathbf{C}} \mathbf{z_j[m]z_j[n]z_k[p]z_k[q]}.$$

Taking expectation and separating the cases $j \neq k$ and $j = k$ yields

$$\mathbb{E}[\mathbf{G_{mn}G_{pq}}] = \frac{1}{C^2} \sum_{\substack{j,k=1 \\ j \neq k}}^{C} \mathbb{E}[\mathbf{z_j[m]z_j[n]z_k[p]z_k[q]}] + \frac{1}{C^2} \sum_{j=1}^{C} \mathbb{E}[\mathbf{z_j[m]z_j[n]z_j[p]z_j[q]}]$$

$$= \frac{C(C-1)}{C^2} \mathbb{E}[\mathbf{z_1[m]z_1[n]}]\,\mathbb{E}[\mathbf{z_2[p]z_2[q]}] + \frac{1}{C}\mathbb{E}[\mathbf{z_1[m]z_1[n]z_1[p]z_1[q]}]$$

$$= \left(1 - \frac{1}{C}\right)\mathbf{M_{mn}M_{pq}} + \frac{1}{C}\mathbf{T_{mnpq}}, \tag{27}$$

where we introduced the fourth-order tensor

$$\mathbf{T_{mnpq}} \overset{\mathrm{def}}{=} \mathbb{E}[\mathbf{z_1[m]z_1[n]z_1[p]z_1[q]}].$$

Equivalently,

$$\mathbb{E}[\mathbf{G_{mn}G_{pq}}] = \mathbf{M_{mn}M_{pq}} + \frac{1}{C}\,\boldsymbol{\Delta}_{\mathbf{mnpq}}, \qquad \boldsymbol{\Delta}_{\mathbf{mnpq}} \overset{\mathrm{def}}{=} \mathbf{T_{mnpq}} - \mathbf{M_{mn}M_{pq}}, \tag{28}$$

so that in tensor form

$$\mathbb{E}[\mathbf{G} \otimes \mathbf{G}] = \mathbf{M} \otimes \mathbf{M} + \frac{1}{C}\,\boldsymbol{\Delta}.$$

Using the linearity of the Frobenius inner product, the quantity that appears in equation (19) can be written as

$$\mathbb{E}\big[\langle \mathbf{G}, \mathbf{M_i}\rangle \mathbf{G}\big] = \langle \mathbf{M}, \mathbf{M_i}\rangle \mathbf{M} + \frac{1}{C}\mathbf{U_i},$$

for some matrix $\mathbf{U_i} \in \mathbb{R}^{\mathbf{(d+1)\times(d+1)}}$ whose entries are linear combinations of the tensor coefficients $\Delta_{mnpq}$. Substituting this expression into equation 28 and combining with equation (21) gives the finite-$C$ gradient

$$\nabla_{\mathbf{M_i}}\mathbf{L_i(M_i)} = \big(\mathbf{2\langle M, M_i\rangle - 2w^\star[i]}\big)\mathbf{M} - \mathbf{2N} + \frac{2}{C}\mathbf{U_i}. \tag{29}$$

Equation (25) corresponds exactly to the leading term in equation 29 when $C \to \infty$, since in that regime $U_i$ is bounded while the $\frac{1}{C}\mathbf{U_i}$ correction vanishes. For any fixed finite $C$, however, the additional term $\frac{2}{C}\mathbf{U_i}$ provides an $\mathcal{O}(1/C)$ perturbation of the gradient. This perturbation can partially cancel the leading term $(\mathbf{2\langle M, M_i\rangle - 2w^\star[i])M - 2N}$ and may create (non-global) stationary points in parameter space, which is consistent with the non-convexity discussion.

**Implications for Theorem 4.** The same finite-$C$ decomposition is also relevant for the proof of Theorem 4 in Appendix B.3. There, the gradients with respect to the parameters $\mathbf{b}$, $\mathbf{a_j}$, $\mathbf{a_{d+1}}$ and $\mathbf{v[j]}$ are expressed in terms of expectations of products involving $G$ (see the expressions preceding the verification that $\mathbf{b} = \begin{bmatrix} -\mathbf{w}^\star \\ 1 \end{bmatrix}$, $\mathbf{a_j} = \begin{bmatrix} \mathbf{e_j} \\ 0 \end{bmatrix}$, $\mathbf{a_{d+1} = 0}$, $\mathbf{v = w^\star}$ form a stationary point in the $C \to \infty$ limit). Using equation 27–equation 28, each such expectation can be written as its infinite-context value plus an $\mathcal{O}(1/C)$ correction. Consequently, for finite $C$ every first-order derivative at $w = w^\star$ takes the form

$$\frac{\partial R(f_{y_q\text{-LSA}})}{\partial \theta} = \underbrace{\frac{\partial R_\infty(f_{y_q\text{-LSA}})}{\partial \theta}}_{=0} + \mathcal{O}\left(\frac{1}{C}\right), \qquad \theta \in \{\mathbf{b}, \mathbf{a_j}, \mathbf{a_{d+1}}, \mathbf{v[j]}\},$$

where $R_\infty$ denotes the risk in the limit $C \to \infty$. Thus, $\mathbf{w = w^\star}$ is an approximate stationary point whose residual gradient decays at rate $\mathcal{O}(1/C)$ as the context size grows. This clarifies how the exact correspondence with one-step gradient descent established in Theorem 4 is approached as $C$ increases, and how small but non-zero biases may appear in practice when $C$ is finite.

### A.4 Proof of Lemma 1

*Proof.* From equation 25, we can compute the Hessian of the function $\mathcal{L}_i(\boldsymbol{M}_i)$, that is,

$$\nabla^2_{\boldsymbol{M}_i} \mathcal{L}_i(\boldsymbol{M}_i) = 2\boldsymbol{M}.$$

We verify that $\boldsymbol{M}$ is positive semi-definite. Indeed, let $\mathbf{u} \in \mathbb{R}^d$ and $u \in \mathbb{R}$. We have

$$
\begin{aligned}
\begin{bmatrix} \mathbf{u}^\top & u \end{bmatrix} \boldsymbol{M} \begin{bmatrix} \mathbf{u} \\ u \end{bmatrix} & \overset{equation\ 22}{=} \begin{bmatrix} \mathbf{u}^\top & u \end{bmatrix} \begin{bmatrix} \boldsymbol{I}_d & \mathbf{w}_\star \\ \mathbf{w}_\star^\top & \|\mathbf{w}_\star\|^2 + d \end{bmatrix} \begin{bmatrix} \mathbf{u} \\ u \end{bmatrix} \\
& = \begin{bmatrix} \mathbf{u}^\top & u \end{bmatrix} \begin{bmatrix} \mathbf{u} + u\mathbf{w}_\star \\ \mathbf{w}_\star^\top \mathbf{u} + u(\|\mathbf{w}_\star\|^2 + d) \end{bmatrix} \\
& = \|\mathbf{u}\|^2 + 2u\mathbf{w}_\star^\top \mathbf{u} + u^2(\|\mathbf{w}_\star\|^2 + d) \\
& = \|\mathbf{u} + u\mathbf{w}_\star\|^2 + du^2 \geq 0.
\end{aligned}
$$

Since $\boldsymbol{M}$ is positive semi-definite, we have the function $\mathcal{L}_i$ is convex with respect to $\boldsymbol{M}_i$.

From equation 18, we know that

$$\mathcal{R}(f_{\mathsf{H-LSA}}) = \sum_{i=1}^{d+1} \mathcal{L}_i(\boldsymbol{M}_i).$$

Each function $\mathcal{L}_i$ is a function of $\boldsymbol{M}_i$. We denote

$$\mathcal{R}(f_{\mathsf{H-LSA}}) = f(\boldsymbol{M}_1, \cdots, \boldsymbol{M}_{d+1}).$$

Then the Hessian of the function $f$ with respect to variables $\boldsymbol{M}_1, \cdots, \boldsymbol{M}_{d+1}$ is a block diagonal matrix, each block on the diagonal is $\nabla^2_{\boldsymbol{M}_i} \mathcal{L}_i(\boldsymbol{M}_i) \geq 0$. Therefore, the function $f$ is convex with respect to $\boldsymbol{M}_1, \cdots, \boldsymbol{M}_{d+1}$.

Lastly, $\boldsymbol{M}_i = \sum_{h=1}^H \mathbf{b}_h(\mathbf{a}_i^h)^\top$ for $i \in [d+1]$. To simplify it, we can consider only one head. That is, $\boldsymbol{M}_i = \mathbf{b}_1(\mathbf{a}_i^1)^\top$, a bilinear function, which is known to be not convex with respect to $\mathbf{b}_1$ and $\mathbf{a}_i^1$.

To conclude, the ICL risk $\mathcal{R}(f_{\mathsf{H-LSA}})$ is a composite function with a convex function and non convex functions, which implies that $\mathcal{R}(f_{\mathsf{H-LSA}})$ is not convex. $\square$

## B  Proofs of Section 4

### B.1  Derivation of equation 8

Here we provide the derivation of equation 8. Recall

$$
\mathbf{W}^V = \begin{bmatrix} 0 & 0 \\ \mathbf{w}_\star^\top & -1 \end{bmatrix}, \quad \mathbf{W}^K = \mathbf{W}^Q = \begin{bmatrix} \boldsymbol{I}_d & 0 \\ 0 & 0 \end{bmatrix}, \quad \mathbf{W}^P = -\frac{\eta}{C} \boldsymbol{I}_{d+1}.
$$

From the standard LSA formulation equation 3 with the given embedding in equation 2, we have

$$
\boldsymbol{K} \overset{def}{=} \boldsymbol{Q} \overset{def}{=} \mathbf{W}^Q \boldsymbol{E} = \begin{bmatrix} \boldsymbol{X}^\top & \mathbf{x}_q \\ 0 & 0 \end{bmatrix},
$$

$$
\boldsymbol{V} \overset{def}{=} \mathbf{W}^V \boldsymbol{E} = \begin{bmatrix} 0 & 0 \\ \mathbf{w}_\star^\top \boldsymbol{X}^\top - \mathbf{y}^\top & \mathbf{w}_\star^\top \mathbf{x}_q - y_q \end{bmatrix}.
$$

So we get the LSA simplified as

$$
f_{\mathsf{LSA}}(\boldsymbol{E}) = \left[ \boldsymbol{E} + \mathbf{W}^P \boldsymbol{V} \mathbf{W}^M \left( \boldsymbol{K}^\top \boldsymbol{Q} \right) \right]_{-1,-1}.
$$

In this case, we have

$$
\boldsymbol{V}\mathbf{W}^M\left(\boldsymbol{K}^\top\boldsymbol{Q}\right) = \begin{bmatrix} 0 & 0 \\ (\mathbf{w}_\star^\top\boldsymbol{X}^\top - \mathbf{y}^\top)\boldsymbol{X}\boldsymbol{X}^\top & (\mathbf{w}_\star^\top\boldsymbol{X}^\top - \mathbf{y}^\top)\boldsymbol{X}\mathbf{x}_q \end{bmatrix},
$$

and LSA recovers the result in Von Oswald et al. (2023), which performs one-step GD on the update of the linear regression parameter initialized at $\mathbf{w}_\star = \mathbf{0}$ with $y_q = 0 = \mathbf{w}_\star^\top\mathbf{x}_q$:

$$
\begin{aligned}
f_{\mathsf{LSA}}(\boldsymbol{E}) &= y_q - \frac{\eta}{C}(\mathbf{w}_\star^\top\boldsymbol{X}^\top - \mathbf{y}^\top)\boldsymbol{X}\mathbf{x}_q \\
&= \left(\mathbf{w}_\star - \frac{\eta}{C}\boldsymbol{X}^\top(\boldsymbol{X}\mathbf{w}_\star - \mathbf{y})\right)^\top\mathbf{x}_q,
\end{aligned}
$$

that yields equation 8.

### B.2 $y_q$-LSA is a Special Case of Linear Transformer Block

In this section, we show that $y_q$-LSA defined in equation 12 is a special case of *linear transformer block* (LTB) presented in Zhang et al. (2024b), which is mentioned in Section 4.

LTB combines LSA with a linear multilayer perceptron (MLP) component. That is,

$$
\begin{aligned}
f_{\mathsf{LTB}} : \mathbb{R}^{(d+1)\times(C+1)} &\to \mathbb{R} \\
\boldsymbol{E} &\mapsto \left[\boldsymbol{W}_2^\top\boldsymbol{W}_1\left(\boldsymbol{E} + \frac{1}{C}\mathbf{W}^P\mathbf{W}^V\boldsymbol{E}\mathbf{W}^M\boldsymbol{E}^\top(\mathbf{W}^K)^\top\mathbf{W}^Q\boldsymbol{E}\right)\right]_{-1,-1},
\end{aligned} \tag{30}
$$

where $\boldsymbol{W}_1, \boldsymbol{W}_2, \mathbf{W}^P, \mathbf{W}^V, \mathbf{W}^K$ and $\mathbf{W}^Q$ are trainable parameters for $f_{\mathsf{LTB}}$, and

$$
\boldsymbol{E} = \begin{bmatrix} \boldsymbol{X}^\top & \mathbf{x}_q \\ \mathbf{y}^\top & 0 \end{bmatrix} \in \mathbb{R}^{(d+1)\times(C+1)},
$$

for $\boldsymbol{X} \in \mathbb{R}^{C\times d}, \mathbf{y} \in \mathbb{R}^C$ and $\mathbf{x}_q \in \mathbb{R}^d$. Notice that there is no initial guess $y_q$ involved in this embedding matrix $\boldsymbol{E}$.

We denote the hypothesis class formed by LTB models as

$$
\mathcal{F}_{\mathsf{LTB}} \stackrel{\text{def}}{=} \left\{ f_{\mathsf{LTB}} : \mathbf{W}^K, \mathbf{W}^Q, \mathbf{W}^V, \mathbf{W}^P, \boldsymbol{W}_1, \boldsymbol{W}_2 \right\},
$$

where $f_{\mathsf{LTB}}$ is defined in equation 30. Then we have the following lemma.

**Lemma 3.** *Consider $f_{y_q-\mathsf{LSA}}$ defined in equation 12. We have*

$$
f_{y_q-\mathsf{LSA}} \in \mathcal{F}_{\mathsf{LTB}}.
$$

*Proof.* Let $\mathbf{w} \in \mathbb{R}^d$. For all $\boldsymbol{X} \in \mathbb{R}^{C\times d}, \mathbf{y} \in \mathbb{R}^C$ and $\mathbf{x}_q \in \mathbb{R}^d$, we have

$$
f_{y_q-\mathsf{LSA}}(\boldsymbol{X}, \mathbf{y}, \mathbf{x}_q) = f_{\mathsf{LSA}}(\boldsymbol{E}_\mathbf{w}) = \left[\boldsymbol{E}_\mathbf{w} + \tfrac{1}{C}\mathbf{W}^P\mathbf{W}^V\boldsymbol{E}_\mathbf{w}\mathbf{W}^M(\boldsymbol{E}_\mathbf{w}^\top(\mathbf{W}^K)^\top\mathbf{W}^Q\boldsymbol{E}_\mathbf{w})\right]_{-1,-1},
$$

with

$$
\boldsymbol{E}_\mathbf{w} = \begin{bmatrix} \boldsymbol{X}^\top & \mathbf{x}_q \\ \mathbf{y}^\top & \mathbf{w}^\top\mathbf{x}_q \end{bmatrix} \in \mathbb{R}^{(d+1)\times(C+1)}.
$$

We aim to find $(\mathbf{W}^K)', (\mathbf{W}^Q)', (\mathbf{W}^V)', (\mathbf{W}^P)', \boldsymbol{W}_1, \boldsymbol{W}_2$ for $f_{\mathsf{LTB}}$ such that $f_{y_q-\mathsf{LSA}}(\boldsymbol{X}, \mathbf{y}, \mathbf{x}_q) = f_{\mathsf{LTB}}(\boldsymbol{E})$ with

$$
\boldsymbol{E} = \begin{bmatrix} \boldsymbol{X}^\top & \mathbf{x}_q \\ \mathbf{y}^\top & 0 \end{bmatrix} \in \mathbb{R}^{(d+1)\times(C+1)}.
$$

Let choose $\boldsymbol{W}_2 = \boldsymbol{I}_{d+1}$ and

$$\boldsymbol{W}_1 = \begin{bmatrix} \boldsymbol{I}_d & \mathbf{w} \\ \mathbf{w}^\top & c \end{bmatrix} \tag{31}$$

with $c \neq \|\mathbf{w}\|^2$, then $\boldsymbol{W}_2^\top \boldsymbol{W}_1 = \boldsymbol{W}_1$ and $\boldsymbol{W}_1 \in \mathbb{R}^{(d+1)\times(d+1)}$ is invertible.

Indeed, let $\mathbf{u} \in \mathbb{R}^d$ and $u \in \mathbb{R}$ such that $\boldsymbol{W}_1 \begin{bmatrix} \mathbf{u} \\ u \end{bmatrix} = 0$. So we have

$$\mathbf{u} + u\mathbf{w} = 0,$$
$$\mathbf{w}^\top \mathbf{u} + cu = 0.$$

From $\mathbf{u} + u\mathbf{w} = 0$, we have $\mathbf{u} = -u\mathbf{w}$. Plugging it into $\mathbf{w}^\top \mathbf{u} + cu = 0$, we obtain

$$(c - \|\mathbf{w}\|^2)u = 0.$$

Since $c \neq \|\mathbf{w}\|^2$, we obtain $u = 0$. Thus, $\mathbf{u} = -u\mathbf{w} = 0$. This implies that $\boldsymbol{W}_1$ is invertible.

Next, we consider the following matrix

$$\boldsymbol{W}_3 = \begin{bmatrix} \boldsymbol{I}_d & 0 \\ \mathbf{w}^\top & 0 \end{bmatrix} \in \mathbb{R}^{(d+1)\times(d+1)}.$$

Let

$$(\mathbf{W}^P)' = \boldsymbol{W}_1^{-1}\mathbf{W}^P,$$
$$(\mathbf{W}^K)' = \mathbf{W}^K \boldsymbol{W}_3,$$
$$(\mathbf{W}^Q)' = \mathbf{W}^Q \boldsymbol{W}_3,$$
$$(\mathbf{W}^V)' = \mathbf{W}^V \boldsymbol{W}_3.$$

We show that $f_{y_q-\mathsf{LSA}}(\boldsymbol{X}, \mathbf{y}, \mathbf{x}_q) = f_{\mathsf{LTB}}(\boldsymbol{E})$.

Indeed, by using $\boldsymbol{X}\mathbf{w} = \mathbf{y}$, we have

$$\boldsymbol{W}_3 \boldsymbol{E} = \begin{bmatrix} \boldsymbol{X}^\top & \mathbf{x}_q \\ \mathbf{w}^\top \boldsymbol{X}^\top & \mathbf{w}^\top \mathbf{x}_q \end{bmatrix} = \boldsymbol{E}_\mathbf{w}.$$

So

$$\begin{aligned}
f_{\mathsf{LTB}}(\boldsymbol{E}) &= \boldsymbol{W}_1 \left[ \left( \boldsymbol{E} + \frac{1}{C} \boldsymbol{W}_1^{-1} \mathbf{W}^P \mathbf{W}^V \boldsymbol{W}_3 \boldsymbol{E} \mathbf{W}^M \boldsymbol{E}^\top (\mathbf{W}^K \boldsymbol{W}_3)^\top \mathbf{W}^Q \boldsymbol{W}_3 \boldsymbol{E} \right) \right]_{-1,-1} \\
&= [\boldsymbol{W}_1 \boldsymbol{E}]_{-1,-1} + \left[ \left( \frac{1}{C} \mathbf{W}^P \mathbf{W}^V \boldsymbol{E}_\mathbf{w} \mathbf{W}^M (\boldsymbol{E}_\mathbf{w}^\top (\mathbf{W}^K)^\top \mathbf{W}^Q \boldsymbol{E}_\mathbf{w}) \right) \right]_{-1,-1} \\
&= \mathbf{w}^\top \mathbf{x}_q + \left[ \left( \frac{1}{C} \mathbf{W}^P \mathbf{W}^V \boldsymbol{E}_\mathbf{w} \mathbf{W}^M (\boldsymbol{E}_\mathbf{w}^\top (\mathbf{W}^K)^\top \mathbf{W}^Q \boldsymbol{E}_\mathbf{w}) \right) \right]_{-1,-1} \\
&= f_{y_q-\mathsf{LSA}}(\boldsymbol{X}, \mathbf{y}, \mathbf{x}_q).
\end{aligned}$$

Thus, we conclude $f_{y_q-\mathsf{LSA}} \in \mathcal{F}_{\mathsf{LTB}}$. $\qquad\square$

### B.3 Proofs of Theorem 4

The risk (loss) function with learnable vector $\mathbf{v}$ is given by:

$$\mathcal{R}(f_{y_q-\mathsf{LSA}}) = \mathbb{E}\left[ \left( \left(\mathbf{E} + \frac{1}{C}\mathrm{Att}(\mathbf{E})\right)_{C+1,C+1} + \mathbf{v}^\top \mathbf{x}_q - \widehat{\mathbf{w}}^T \mathbf{x}_q \right)^2 \right].$$

Similar as Appendix A, we rewrite the risk:

$$
\begin{aligned}
\mathcal{R}(f_{y_q-\mathsf{LSA}}) &= \mathbb{E}\left[\left((1 + \mathbf{b}^T\mathbf{G}\mathbf{a}_{d+1})y_q + (\mathbf{b}^T\mathbf{G}\mathbf{A}_{:d} - \widehat{\mathbf{w}}^\top)\mathbf{x}_q\right)^2\right] \\
&= \mathbb{E}\left[\left((1 + \mathbf{b}^T\mathbf{G}\mathbf{a}_{d+1})\mathbf{v}^\top + (\mathbf{b}^T\mathbf{G}\mathbf{A}_{:d} - \widehat{\mathbf{w}}^\top)\right)\mathbf{x}_q\right] \\
&= \mathbb{E}\left[\sum_{j=1}^{d}\left(\langle\mathbf{G}, \mathbf{b}\mathbf{a}_j^\top\rangle + \langle\mathbf{G}, \mathbf{b}\mathbf{a}_{d+1}^\top\rangle\mathbf{v}[j] + \mathbf{v}[j] - \widehat{\mathbf{w}}[j]\right)^2\right]
\end{aligned}
$$

We define, for each $j$:

$$
t_j = \langle\mathbf{G}, \mathbf{b}\,\mathbf{a}_j^\top\rangle + \langle G, \mathbf{b}\,\mathbf{a}_{d+1}^\top\rangle\mathbf{v}[j] + \mathbf{v}[j] - \widehat{\mathbf{w}}[j].
$$

Then

$$
f_{y_q-\mathsf{LSA}} = \sum_{j=1}^{d}\mathbb{E}\left[t_j^2\right].
$$

**Step 1: Gradient for parameters**

We list the first-order partial derivatives with respect to $\mathbf{b}, \mathbf{a}_j, \mathbf{a}_{d+1}$, and $\mathbf{v}[j]$. $j$ is from 1 to $d$

- **Gradient w.r.t. b**

$$
\frac{\partial t_j}{\partial\mathbf{b}} = \mathbf{G}\,\mathbf{a}_j + \mathbf{v}[j]\,\mathbf{G}\,\mathbf{a}_{d+1}.
$$

$$
\frac{\partial}{\partial\mathbf{b}}\left(t_j^2\right) = 2\,t_j\,\frac{\partial t_j}{\partial\mathbf{b}} = 2\,t_j\left(\mathbf{G}\,\mathbf{a}_j + \mathbf{v}[j]\,\mathbf{G}\,\mathbf{a}_{d+1}\right).
$$

$$
\frac{\partial f}{\partial\mathbf{b}} = \sum_{j=1}^{d}\mathbb{E}\left[2\,t_j\left(\mathbf{G}\,\mathbf{a}_j + \mathbf{v}[j]\,\mathbf{G}\,\mathbf{a}_{d+1}\right)\right].
$$

- **Gradient w.r.t. $\mathbf{a}_j$**

$$
\frac{\partial t_j}{\partial\mathbf{a}_j} = \mathbf{G}^\top\mathbf{b}.
$$

$$
\frac{\partial}{\partial\mathbf{a}_j}\left(t_j^2\right) = 2\,t_j\left(\mathbf{G}^\top\mathbf{b}\right).
$$

Only the $j$-th term depends on $\mathbf{a}_j$, so

$$
\frac{\partial f_{y_q-\mathsf{LSA}}}{\partial\mathbf{a}_j} = 2\,\mathbb{E}\left[t_j\left(\mathbf{G}^\top\mathbf{b}\right)\right].
$$

- **Gradient w.r.t. $\mathbf{a}_{d+1}$**

$$
\frac{\partial t_j}{\partial\mathbf{a}_{d+1}} = \mathbf{v}[j]\left(\mathbf{G}^\top\mathbf{b}\right).
$$

$$
\frac{\partial}{\partial\mathbf{a}_{d+1}}\left(t_j^2\right) = 2\,t_j\left(\mathbf{v}[j]\,\mathbf{G}^\top\mathbf{b}\right).
$$

$$\frac{\partial f_{y_q-\mathsf{LSA}}}{\partial \mathbf{a}_{d+1}} = 2\sum_{j=1}^{d} \mathbb{E}\Big[t_j\,\mathbf{v}[j]\,(\mathbf{G}^\top \mathbf{b})\Big].$$

- **Gradient w.r.t. $v[j]$**

We have

$$t_j = \mathbf{b}^\top \mathbf{G}\,\mathbf{a}_j + v[j]\left(\mathbf{b}^\top \mathbf{G}\,\mathbf{a}_{d+1}+1\right) - \left(\mathbf{w}[j]+\mathbf{w}_\star[j]\right).$$

$$\frac{\partial t_j}{\partial v[j]} = \left(\mathbf{b}^\top \mathbf{G}\,\mathbf{a}_{d+1}+1\right).$$

$$\frac{\partial f_{y_q-\mathsf{LSA}}}{\partial v[j]} = 2\,\mathbb{E}\Big[t_j\left(\mathbf{b}^\top \mathbf{G}\,\mathbf{a}_{d+1}+1\right)\Big].$$

**Step 2: Plug in One Step GD**

we verify when $\mathbf{b} = \begin{bmatrix} -\mathbf{w}_\star \\ 1 \end{bmatrix}$, $\mathbf{a}_j = \begin{bmatrix} \mathbf{e}_j \\ 0 \end{bmatrix}$, $\mathbf{a}_{d+1}=0$, $\mathbf{v}=\mathbf{w}_\star$, the gradients equal to zero

we define $\mathbf{w} = \widehat{\mathbf{w}} - \mathbf{w}_\star$, We have the following intermediate formula:

$$\mathbf{b}^T \mathbf{G}\mathbf{a}_j = [-\mathbf{w}_\star^T, 1]\sum_{i=1}^{C} \begin{bmatrix} \mathbf{x}_i\mathbf{x}_i^T & \mathbf{x}_i y_i^T \\ y_i\mathbf{x}_i^T & y_i^2 \end{bmatrix}\begin{bmatrix} \mathbf{e}_j \\ 0 \end{bmatrix} = \frac{\sum_{i=1}^{C}}{C}\left[\mathbf{w}^T\mathbf{x}_i\mathbf{x}_i^T, \mathbf{w}^T\mathbf{x}_i y_i\right]\left[\begin{bmatrix} \mathbf{e}_j \\ 0 \end{bmatrix}\right] = \frac{\sum_{i=1}^{C}}{C}\mathbf{w}^T\mathbf{x}_i\mathbf{x}_i[j]$$

$$v[i](\mathbf{b}^T \mathbf{G}\mathbf{a}_{d+1}) = 0$$

$$t_j = \frac{\sum_{i=1}^{C}}{C}\mathbf{w}^T\mathbf{x}_i\mathbf{x}_i[j] - \mathbf{w}[j]$$

$$\mathbf{G}a_j = \frac{1}{C}\sum_{i=1}^{C} \begin{bmatrix} \mathbf{x}_i\mathbf{x}_i[j] \\ y_i\mathbf{x}_i[j] \end{bmatrix}$$

- **Gradient w.r.t. b**

$$\frac{\partial f_{y_q-\mathsf{LSA}}}{\partial b} = 2\sum_{j=1}^{d} \mathbb{E}\left[\left(\frac{\sum_{i=1}^{C}}{C}\mathbf{w}^T\mathbf{x}_i\mathbf{x}_i[j] - \mathbf{w}[j]\right)\frac{1}{C}\sum_{i=1}^{C}\begin{bmatrix} \mathbf{x}_i\mathbf{x}_i[j] \\ y_i\mathbf{x}_i[j] \end{bmatrix}\right]$$

Calculate each part:

$$-\mathbf{w}[j]\frac{1}{C}\sum_{i=1}^{C}\mathbf{x}_i\mathbf{x}_i[j] = 0,$$

$$-\mathbf{w}[j]\frac{1}{C}\sum_{i=1}^{C}y_i\mathbf{x}_i[j] = -\mathbf{w}[j]\frac{1}{C}\sum_{i=1}^{C}(\mathbf{w}_\star^T + \mathbf{w}^T)\mathbf{x}_i\mathbf{x}_i[j] = -\mathbf{w}[j]\frac{1}{C}\sum_{i=1}^{C}\mathbf{w}^T\mathbf{x}_i\mathbf{x}_i[j] = -1,$$

$$\frac{\sum_{i=1}^{C}}{C}\mathbf{w}^T\mathbf{x}_i\mathbf{x}_i[j]\mathbf{x}_i\mathbf{x}_i[j] = 0,$$

$$\frac{\sum_{i=1}^{C} \mathbf{w}^T \mathbf{x}_i \mathbf{x}_i[j]}{C} \frac{1}{C} \sum_{i=1}^{C} (\mathbf{w}_\star^T + \mathbf{w}^T) \mathbf{x}_i \mathbf{x}_i[j] = \frac{\sum_{i=1}^{C} \mathbf{w}^T \mathbf{x}_i \mathbf{x}_i[j]}{C} \frac{\sum_{i=1}^{C} \mathbf{w}^T \mathbf{x}_i \mathbf{x}_i[j]}{C}$$

$$= \frac{1}{C^2} \mathbb{E}\left[ \left( \sum_{i=1}^{C} \mathbf{x}_i[j]\, \mathbf{x}_i^T \right) \left( \sum_{k=1}^{C} \mathbf{x}_k[j]\, \mathbf{x}_k \right) \right],$$

**compute** $\mathbb{E}\left[ \left( \sum_{i=1}^{C} \mathbf{x}_i[j]\, \mathbf{x}_i^T \right) \left( \sum_{k=1}^{C} \mathbf{x}_k[j]\, \mathbf{x}_k \right) \right]$

when $i \neq k$ ,

$$\mathbb{E}[\mathbf{x}_i[j]\mathbf{x}_k[j](\mathbf{x}_i^T \mathbf{x}_k)] = 1.$$

$$\sum_{i \neq k} \mathbb{E}[\mathbf{x}_i[j]\mathbf{x}_k[j](\mathbf{x}_i^T \mathbf{x}_k)] = C(C-1) \cdot 1 = C(C-1).$$

when $i = k$,

$$\mathbf{x}_i[j]\mathbf{x}_i[j](\mathbf{x}_i^T \mathbf{x}_i) = \mathbf{x}_i[j]^2 \sum_{m=1}^{d} \mathbf{x}_i[m]^2 = \mathbf{x}_i[j]^2 (\mathbf{x}_i^T \mathbf{x}_i) = d + 2.$$

Because $\mathbb{E}[\mathbf{x}[j]^2(\mathbf{x}^T\mathbf{x})] = \mathbb{E}[\mathbf{x}[j]^4] + \sum_{m \neq j} \mathbb{E}[\mathbf{x}[j]^2\mathbf{x}[m]^2]$ , $\mathbb{E}[\mathbf{x}[j]^4] = 3$.

$$\left( \sum_{i=1}^{C} \mathbf{x}_i[j]\, \mathbf{x}_i^T \right) \left( \sum_{k=1}^{C} \mathbf{x}_k[j]\, \mathbf{x}_k \right) = C(C-1) + C(d+2)$$

if we have very large $C$, we have:

$$\frac{\sum_{i=1}^{C} \mathbf{w}^T \mathbf{x}_i \mathbf{x}_i[j]}{C} \frac{1}{C} \sum_{i=1}^{C} y_i \mathbf{x}_i[j] = 1.$$

So that $\frac{\partial f_{y_q - \mathsf{LSA}}}{\partial \mathbf{b}} = 0$

- **Gradient w.r.t. $\mathbf{a}_j$**

$$\frac{\partial f_{y_q - \mathsf{LSA}}}{\partial \mathbf{a}_j} = 2\, \mathbb{E}\big[ t_j \left( \mathbf{G}^\top \mathbf{b} \right) \big] = \mathbb{E}\big[ \frac{\sum_{i=1}^{C} \mathbf{w}^T \mathbf{x}_i \mathbf{x}_i[j]}{C} - \mathbf{w}[j]) \frac{\sum_{i=1}^{C}}{C} \begin{bmatrix} \mathbf{x}_i \mathbf{x}_i^T \mathbf{w} \\ y_i \mathbf{x}_i^T \mathbf{w} \end{bmatrix} \big]$$

$$\mathbb{E}\big[ -\mathbf{w}[j] \frac{\sum_{i=1}^{C}}{C} \begin{bmatrix} \mathbf{x}_i \mathbf{x}_i^T \mathbf{w} \\ (\mathbf{w}^T + \mathbf{w}_\star^T)\mathbf{x}_i \mathbf{x}_i^T \mathbf{w} \end{bmatrix} \big] = \mathbb{E}\big[ - \begin{bmatrix} \mathbf{e}_j \\ \mathbf{w}[j](\mathbf{w}^T + \mathbf{w}_\star^T)\mathbf{w} \end{bmatrix} \big] = - \begin{bmatrix} \mathbf{e}_j \\ \mathbf{w}_\star[j] \end{bmatrix}$$

**compute** $\frac{\sum_{i=1}^{C} \mathbf{w}^T \mathbf{x}_i \mathbf{x}_i[j]}{C} \frac{\sum_{i=1}^{C} \mathbf{x}_i \mathbf{x}_i^T \mathbf{w}}{C}$

We aim to compute the expectation:

$$\mathbb{E}\left[ \mathbf{w}^T \left( \sum_{i=1}^{C} \mathbf{x}_i \mathbf{x}_i[j] \right) \left( \sum_{k=1}^{C} \mathbf{x}_k \mathbf{x}_k^T \right) \mathbf{w} \right],$$

First, expand the product inside the expectation:

$$\mathbf{w}^T \left( \sum_{i=1}^{C} \mathbf{x}_i \mathbf{x}_i[j] \right) \left( \sum_{k=1}^{C} \mathbf{x}_k \mathbf{x}_k^T \right) \mathbf{w} = \sum_{i=1}^{C} \sum_{k=1}^{C} \mathbf{w}^T \mathbf{x}_i \, \mathbf{x}_i[j] \, \mathbf{x}_k^T \mathbf{w} \cdot \mathbf{x}_k.$$

Taking expectation:

$$\mathbb{E} \left[ \sum_{i=1}^{C} \sum_{k=1}^{C} \mathbf{w}^T \mathbf{x}_i \, \mathbf{x}_i[j] \, \mathbf{x}_k^T \mathbf{w} \cdot \mathbf{x}_k \right] = \sum_{i=1}^{C} \sum_{k=1}^{C} \mathbb{E} \left[ \mathbf{w}^T \mathbf{x}_i \, \mathbf{x}_i[j] \, \mathbf{x}_k^T \mathbf{w} \cdot \mathbf{x}_k \right].$$

Case 1: $i \neq k$

Since $\mathbf{x}_i$ and $\mathbf{x}_k$ are independent:

$$\mathbb{E} \left[ \mathbf{w}^T \mathbf{x}_i \, \mathbf{x}_i[j] \, \mathbf{x}_k^T \mathbf{w} \cdot \mathbf{x}_k \right] = \mathbb{E} \left[ \mathbf{w}^T \mathbf{x}_i \, \mathbf{x}_i[j] \right] \mathbb{E} \left[ \mathbf{x}_k^T \mathbf{w} \cdot \mathbf{x}_k \right].$$

Given $\mathbf{x}_i \sim \mathcal{N}(0, I_d)$:

$$\mathbb{E} \left[ \mathbf{w}^T \mathbf{x}_i \, \mathbf{x}_i[j] \right] = \mathbf{w}[j], \quad \mathbb{E} \left[ \mathbf{x}_k^T w \cdot \mathbf{x}_k \right] = \mathbf{w}.$$

Thus, for $i \neq k$:

$$\mathbb{E} \left[ \mathbf{w}^T \mathbf{x}_i \, \mathbf{x}_i[j] \, \mathbf{x}_k^T \mathbf{w} \cdot \mathbf{x}_k \right] = \mathbf{w}[j] \mathbf{w}.$$

There are $C(C-1)$ such terms, contributing:

$$C(C-1) \mathbf{w}[j] w = C(C-1) e_j.$$

Case 2: $i = k$

For $i = k$:

$$\mathbb{E} \left[ \mathbf{w}^T \mathbf{x}_i \, \mathbf{x}_i[j] \, \mathbf{x}_i^T w \cdot \mathbf{x}_i \right] = \mathbb{E} \left[ (\mathbf{w}^T \mathbf{x}_i)^2 \mathbf{x}_i[j] \mathbf{x}_i \right].$$

Using properties of Gaussian vectors:

$$\mathbb{E} \left[ (\mathbf{w}^T \mathbf{x}_i)^2 \mathbf{x}_i[j] \mathbf{x}_i \right] = 2 \mathbf{w}_j \mathbf{w} + \|w\|^2 \mathbf{e}_j,$$

where $e_j$ is the $j$-th standard basis vector. There are $C$ such terms, contributing:

$$C(2 \mathbf{w}_j w + \|\mathbf{w}\|^2 \mathbf{e}_j).$$

Adding contributions from both cases:

$$\mathbb{E} \left[ \mathbf{w}^T \left( \sum_{i=1}^{C} \mathbf{x}_i \mathbf{x}_i[j] \right) \left( \sum_{k=1}^{C} \mathbf{x}_k \mathbf{x}_k^T \right) w \right] = C(C-1) \mathbf{w}_j \|\mathbf{w}\|^2 + C(2 \mathbf{w}_j \mathbf{w} + \|\mathbf{w}\|^2 \mathbf{e}_j).$$

Simplifying:

$$= C(C+1) \mathbf{w}_j \mathbf{w} + C \|\mathbf{w}\|^2 \mathbf{e}_j.$$

Thus, the expectation is:

$$\mathbb{E} \left[ \mathbf{w}^T \left( \sum_{i=1}^{C} \mathbf{x}_i \mathbf{x}_i[j] \right) \left( \sum_{k=1}^{C} \mathbf{x}_k \mathbf{x}_k^T \right) \mathbf{w} \right] = C(C+1) \mathbf{w}_j \mathbf{w} + C \|\mathbf{w}\|^2 \mathbf{e}_j.$$

when $C$ is large $\mathbb{E} \left[ \frac{\sum_{i=1}^{C} \mathbf{w}^T \mathbf{x}_i \mathbf{x}_i[j]}{C} \frac{\sum_{i=1}^{C} \mathbf{x}_i \mathbf{x}_i^T \mathbf{w}}{C} \right] = \mathbf{e}_j$

**compute** $\frac{\sum_{i=1}^{C} \mathbf{w}^T \mathbf{x}_i \mathbf{x}_i[j]}{C} \frac{\sum_{i=1}^{C} y_i \mathbf{x}_i^T}{C} w$

$$\frac{\sum_{i=1}^{C} \mathbf{w}^T \mathbf{x}_i \mathbf{x}_i[j]}{C} \frac{\sum_{k=1}^{C} (\mathbf{w}_\star^T + \mathbf{w}^T) \mathbf{x}_k \mathbf{x}_k^T}{C} w$$

From our previous experience, we only need calculate case when $k \neq i$

$$\mathbb{E}\left[ \frac{\sum_{i=1}^{C} \mathbf{w}^T \mathbf{x}_i \mathbf{x}_i[j]}{C} \frac{\sum_{k=1}^{C} (\mathbf{w}_\star^T + \mathbf{w}^T) \mathbf{x}_k \mathbf{x}_k^T}{C} w \right] = \mathbb{E}\left[ \mathbf{w}[j] (\mathbf{w}_\star^T + \mathbf{w}^T) w \right] = \mathbf{w}_\star[j]$$

So that we have $\frac{\partial f_{y_q - \mathsf{LSA}}}{\partial \mathbf{a}_j} = 0$

- **Gradient w.r.t. $\mathbf{a}_{d+1}$**

$$\frac{\partial f_{y_q - \mathsf{LSA}}}{\partial \mathbf{a}_{d+1}} = 2 \sum_{j=1}^{d} \mathbb{E}\left[ t_j \, \mathbf{v}[j] \, (\mathbf{G}^\top \mathbf{b}) \right] = 2 \sum_{j=1}^{d} \mathbb{E}\left[ t_j \, \mathbf{w}_\star[j] \, (\mathbf{G}^\top \mathbf{b}) \right].$$

we already have $\frac{\partial f_{y_q - \mathsf{LSA}}}{\partial \mathbf{a}_j} = 2 \mathbb{E}\left[ t_j \, (\mathbf{G}^\top \mathbf{b}) \right]$

So that we have $\frac{\partial f_{y_q - \mathsf{LSA}}}{\partial \mathbf{a}_{d+1}} = 0$

- **Gradient w.r.t. $v[j]$**

$$\frac{\partial f_{y_q - \mathsf{LSA}}}{\partial v[j]} = 2 \mathbb{E}\left[ t_j \left( \mathbf{b}^\top \mathbf{G} \, \mathbf{a}_{d+1} + 1 \right) \right] = 2 \mathbb{E}\left[ \frac{(\sum_{i=1}^{C} \mathbf{w}^T \mathbf{x}_i \mathbf{x}_i[j] - \mathbf{w}[j])}{C} \left( \frac{\sum_{i=1}^{C} \mathbf{w}^T \mathbf{x}_i \mathbf{x}_i[j] + 1}{C} \right) \right]$$

$$2 \mathbb{E}\left[ \frac{(\sum_{i=1}^{C} \mathbf{w}^T \mathbf{x}_i \mathbf{x}_i[j] - \mathbf{w}[j])}{C} 1 \right] = 0$$

$$2 \mathbb{E}\left[ \frac{(\sum_{i=1}^{C} \mathbf{w}^T \mathbf{x}_i \mathbf{x}_i[j] - \mathbf{w}[j])}{C} \frac{\sum_{k=1}^{C} \mathbf{w}^T \mathbf{x}_k \mathbf{x}_k[j]}{C} \right]$$

we still only consider the case $i \neq k$

$$2 \mathbb{E}\left[ \frac{(\sum_{i=1}^{C} \mathbf{w}^T \mathbf{x}_i \mathbf{x}_i[j] - \mathbf{w}[j])}{C} \frac{\sum_{k=1}^{C} \mathbf{w}^T \mathbf{x}_k \mathbf{x}_k[j]}{C} \right] = 2 \mathbb{E}\left[ \mathbf{w}[j] - \mathbf{w}[j]) \mathbf{w}^T \right] = 0$$

**we verify that $\mathbf{b} = \begin{bmatrix} -\mathbf{w}_\star \\ 1 \end{bmatrix}$, $\mathbf{a}_j = \begin{bmatrix} \mathbf{e}_j \\ 0 \end{bmatrix} \mathbf{a}_{d+1} = 0 \; v = \mathbf{w}_\star$, is a stationary point for loss $f_{y_q - \mathsf{LSA}}$**

### B.4 Proof of Lemma 2

*Proof.* Based on the proof of Lemma 3, we consider the following matrix

$$\boldsymbol{W} = \begin{bmatrix} \boldsymbol{I}_d & 0 \\ \mathbf{w}^\top & 0 \end{bmatrix} \in \mathbb{R}^{(d+1) \times (d+1)}.$$

Now for any $f_{y_q - \mathsf{LSA}}$'s inputs $(\boldsymbol{X}, \mathbf{y}, \mathbf{x}_q)$, by using $\boldsymbol{X}\mathbf{w} = \mathbf{y}$, we have

$$\boldsymbol{W}\boldsymbol{E} = \begin{bmatrix} \boldsymbol{X}^\top & \mathbf{x}_q \\ \mathbf{w}^\top \boldsymbol{X}^\top & \mathbf{w}^\top \mathbf{x}_q \end{bmatrix} = \boldsymbol{E}_{\mathbf{w}}.$$

Thus,

$$f_{y_q-\mathsf{LSA}}(\boldsymbol{X}, \mathbf{y}, \mathbf{x}_q) = f_{\mathsf{LSA}}(\boldsymbol{E}_{\mathbf{w}}) = f_{\mathsf{LSA}}(\boldsymbol{WE}).$$

By using Lemma 1 with one-single head, we know that $\mathcal{R}(f_{\mathsf{LSA}})$ is non-convex. Thus, we conclude that $\mathcal{R}(f_{y_q-\mathsf{LSA}})$ is non-convex, as it is a composite function with a non-convex function $\mathcal{R}(f_{\mathsf{LSA}})$ and a linear function. $\qquad\square$

## C  Details of Experiment

### C.1  Implementation Settings.

The experiments use JAX to implement and train the LSA models. We set the learning rate to $lr = 5 \times 10^{-4}$ and a batch size of 2,048. A single linear attention layer is used, without any LayerNorm or softmax operations. We will release our code repository upon publication to facilitate reproducibility.

Table 1: Overview of the experimental setups. Each experiment modifies one factor (number of attention heads, prior mean, or $y_q$) while holding the others fixed.

| Experiment | Number of Heads | Prior Mean | $y_q$ |
|---|---|---|---|
| Head Section 5.1.1 | Varies | $[2, 2, \ldots, 2]$ | 0 |
| Prior Mean Section 5.1.2 | 11 | Varies | 0 |
| $y_q$ Section 5.1.3 | 11 | $[0, 0, \ldots, 0]$ | Varies |

### C.2  Detailed Metric Definitions

**Prediction Norm Difference** The *prediction norm difference* measures the discrepancy between the outputs of $y_q$-LSA and one-step GD ($f_{GD}$). Given a test input $\mathbf{x}_q$, we define the difference as:

$$\|f_{y_q-\mathsf{LSA}}(\mathbf{x}_q) - f_{GD}(\mathbf{x}_q)\|.$$

This metric quantifies how closely $y_q$-LSA approximates the predictions of the explicit one-step GD solution.

**Gradient Norm Difference** The *gradient norm difference* assesses the deviation between the sensitivity of the model predictions to the input. Given the gradient of the output with respect to the input $\mathbf{x}_q$, we compute:

$$\left\| \frac{\partial f_{GD}(\mathbf{x}_q)}{\partial \mathbf{x}_q} - \frac{\partial f_{y_q-\mathsf{LSA}}(\mathbf{x}_q)}{\partial \mathbf{x}_q} \right\|.$$

This metric evaluates whether $y_q$-LSA captures the same local sensitivity as one-step GD.

**Cosine Similarity** The *cosine similarity* measures the angular alignment between the gradients of the two models. It is defined as:

$$\frac{\left\langle \frac{\partial f_{GD}(\mathbf{x}_q)}{\partial \mathbf{x}_q}, \frac{\partial f_{y_q-\mathsf{LSA}}(\mathbf{x}_q)}{\partial \mathbf{x}_q} \right\rangle}{\left\| \frac{\partial f_{GD}(\mathbf{x}_q)}{\partial \mathbf{x}_q} \right\| \left\| \frac{\partial f_{y_q-\mathsf{LSA}}(\mathbf{x}_q)}{\partial \mathbf{x}_q} \right\|}.$$

A cosine similarity of 1 indicates perfect alignment between the two models, while lower values suggest deviations in the learned representations.

### C.3  LLM Experimental Settings

We conducted our experiments using the STS-Benchmark dataset (English subset)(May, 2021), which consists of sentence pairs labelled with semantic similarity scores ranging from 0 to 5. The LLM used in our study was Meta-LLaMA-3.1-8B-Instruct(Grattafiori et al., 2024) and Qwen/Qwen2.5-7B-Instruct(Yang et al., 2024;

Team, 2024). The model's generation parameters included a maximum of 150 new tokens and deterministic decoding.

The guess model was trained to generate initial similarity score guesses. It consisted of a two-layer feedforward architecture, taking as input the concatenated embeddings of two sentences computed by the Sentence-Transformer model all-MiniLM-L6-v2(Reimers & Gurevych, 2020). The first layer mapped the concatenated embeddings to a 16-dimensional space with ReLU activation, followed by a second layer that outputs a single scalar value as the predicted similarity score. The model was trained using Adam Optimizer(Kingma, 2014) with a learning rate of 1e-3 and a mean squared error loss function. Training was performed over 10 epochs, with a batch size of 8. Sentence embeddings were dynamically computed during training. The loss for training the guess model was computed as the MSE between the predicted and ground truth scores.

For each prompt, a context was constructed by randomly sampling 10 labelled examples from the dataset. Each labelled example included two sentences, a ground truth similarity score, and an initial guess for the similarity score generated by a lightweight guess model. The query example included two sentences and its guessed similarity score and an explicit instruction for the LLM to refine the guess and provide a similarity score between 0 and 5.

To evaluate the effectiveness of the initial guess, we calculated the MSE between the LLM's predicted similarity scores and the ground truth scores across 100 experimental runs. The baseline performance, derived from the initial guesses provided was compared to the refined predictions generated by the LLM.

