# OpenReview forum: "The Initialization Determines Whether In-Context Learning Is Gradient Descent"
_TMLR — Accepted by TMLR_

### Review · Reviewer_2Dxp · 2025-10-14

**Summary Of Contributions:**

This paper studies the setting of in-context  linear regression with linear self-attention (LSA) where the underlying parameter $\hat{w} \sim N(w^* , I_d)$ with $w^* \neq 0$. Previous work [1] has shown that when $w^* = 0$ the global minimizer of one layer LSA is one step of pre-conditioned gradient descent. In this paper, the authors focus on the case of non-zero $w^* $ and show how in this case the loss is non-convex and thus does not have a global minimizer. They then introduce a learnable parameter in the input $w$, and when $w = w^*$ the LSA is in fact one step of gradient descent for specific choice. They then perform experiments to show the gap of one step of gradient descent and the trained model, while also exploring this as a function of the number of heads. They also perform some experiments in language models where they add some extra initial guesses in the prompt (these values are attained from another smaller model).

[1]: Ahn, Kwangjun, et al. "Transformers learn to implement preconditioned gradient descent for in-context learning." Advances in Neural Information Processing Systems 36 (2023): 45614-45650.

**Audience:**

Yes

**Audience Explanation:**

I think this work provides some understanding on how even changing the initialization affects what these models are learning.

**Broader Impact Concerns:**

No ethical concerns.

**Claims And Evidence:**

Yes

**Claims Explanation:**

Most of the claims are clearly stated.

**Requested Changes:**

Regarding the experimental section in figure 5 what is the GD loss correspond to? Since we have different initializations for w, the loss of one GD step should be different. I would like the authors to clarify what exactly this GD loss is. I would also like a clarification paragraph about the differences between $C \to \infty$ and $C$ being a constant. How the results would be affected?

---

> ### Author Response · Authors · 2025-10-28
> **Author Response to Reviewer 2Dxp**
>
> We thank the reviewer for the careful read and for the constructive requests for clarification. Below we address the two main points we will make in the manuscript.
>
> #### “What exactly is the GD loss in Figure 5? Since we have different initializations for (w), the loss of one GD step should be different.”
>
> In Figure 5, the dashed “one-step GD” line is the in-context risk (MSE on the query) obtained by one step of gradient descent on the least-squares objective built from the context pairs ((X,y)), starting from the prior mean initialization ($w_0 = w_\star$). The update and the resulting prediction are:
> $$ w_1 = w_0 - \frac{\eta}{C}X^\top(Xw_0 - y),\qquad
> \hat y_{\text{GD}} = x_q^\top w_1, $$
> and the plotted baseline is
> $R_{\mathrm{GD-1step}} = \mathbb{E}[({\hat{y}}_{\mathrm{GD}} - y)^2]$
>
> In earlier analyses such as von Oswald et al. the GD formulation assumes zero initialization, i.e. $w_0 = 0$, together with a zero-mean Gaussian prior on $w_\star$. Under these assumptions, the least-squares objective $w_1 = \frac{\eta}{C} X^\top y,$
>
> [1] Ahn, Kwangjun, et al. "Transformers learn to implement preconditioned gradient descent for in-context learning." Advances in Neural Information Processing Systems 36.
>
> ---
> #### “Please add a clarification paragraph about the differences between $C \to \infty$ and $C$ being a constant. How would the results be affected?”
>
>
> In Theorem 2 Proof we explicitly use the identity
> $$\mathbb{E}[G_{mn}G_{pq}] = \frac{C(C-1)}{C^2}\mathbb{E}[z_1[m]z_1[n]]\mathbb{E}[z_2[p]z_2[q]] + \frac{1}{C}\mathbb{E}[z_1[m]z_1[n]z_1[p]z_1[q]].$$
>
> Hence the fourth-order term is of order $1/C$ and vanishes as $C \to \infty$, yielding $\mathbb{E}[G \otimes G] = M \otimes M$ and the closed-form gradient $(2\langle M\, M_i \rangle - 2w_\star [i])M - 2N$ used to show no non-trivial stationary point when $w_\star \neq 0$. For finite $C$ we obtain an extra $\mathcal{O}(1/C)$ correction in the gradient, which may permit stationary points, consistent with our discussion on non-convexity.
>
> In Theorem 4，
> for yq-LSA with $w = w_\star$ and weights in Eq.(7), all first-order derivatives vanish in the $C \to \infty$ limit. With finite $C$, every gradient component equals its infinite-$C$ value plus an $\mathcal{O}(1/C)$ term coming from $\mathbb{E}[G \otimes G] - M \otimes M$. Thus $w = w_\star$ is an approximate stationary point with bias decaying as $1/C$.
>
>
> All corresponding clarifications have been incorporated into the revised manuscript, highlighted in dark blue for clarity. We thank the reviewer again for the valuable suggestion。

---

> > ### Comment · Reviewer_2Dxp · 2025-11-17
> > **Reply to authors**
> >
> > I would like to thank the authors for the response and for adding the discussion on the case of $C$ finite.
> >
> > Regarding the first point. My question is why the GD loss after one step is exactly the same while $w^* =w_0$ changes? You have 3 different values: $w^* =[1,1,..,1], w^* =[2,2,..,2], w^* =[3,3,..,3]$. I would expect the iterations to not be exactly the same when the algorithm starts from different initializations.

---

> > > ### Author Response · Authors · 2025-11-18
> > >
> > > Thank you for the follow-up question. We are happy to clarify why the one-step GD baseline in Figure 5 is identical across the prior mean $w^\star$.
> > >
> > >
> > > Recall the generative model in Section 2: for a fixed prior mean $w^\star\in\mathbb{R}^d$, a task parameter is drawn as $\hat w \sim \mathcal N(w^\star, I_d),$ the features $x_q,x_1,\dots,x_C$ are i.i.d. $\mathcal N(0,I_d)$, and the labels are $y = \langle \hat w, x_q\rangle,  y_i = \langle \hat w, x_i\rangle,\ i\in[C].$
> > > With step size $\eta$ and initialization $w_0 = w^\star$, one step of GD on the least-squares loss gives the iterate $w_1 = w^\star - \frac{\eta}{C} X^\top (X w^\star - {\bf y}),$ so Eq. (10) can be written as $f_{\mathrm{GD}}(E) = \langle w_1, x_q\rangle .$
> > >
> > > In Figure 5, the dashed curve corresponds to the one-step GD in-context risk
> > > $$
> > > R(f_{\mathrm{GD}})=\mathbb{E}\left[(f_{\mathrm{GD}}(E)-\mathbf{y})^2\right],
> > > $$
> > > where $f_{\mathrm{GD}}(E)$ is the prediction obtained by applying one gradient descent update to the least squares objective built from the context pairs $(\mathbf{x}_i,y_i)$, starting from the prior mean $w^\star$ as initialization.
> > >
> > > Using ${\bf y} = X \hat w$ and $\hat w = w^\star + \epsilon$ with $\epsilon \sim \mathcal N(0,I_d)$, we obtain
> > > $$
> > > w_1 - \hat w
> > > = - \frac{\eta}{C} X^\top (X w^\star - {\bf y}) - \epsilon
> > > = - \frac{\eta}{C} X^\top X (w^\star - \hat{w}) - \epsilon
> > > = \Bigl(\tfrac{\eta}{C} X^\top X - I_d\Bigr)\epsilon,
> > > $$
> > > and hence the prediction error on the query is
> > > $$
> > > f_{\mathrm{GD}}(E) - y
> > > = x_q^\top (w_1 - \hat w)
> > > = x_q^\top\Bigl(\tfrac{\eta}{C} X^\top X - I_d\Bigr)\epsilon.
> > > $$
> > > Therefore the one-step GD ICL risk is
> > > $$
> > > R(f_{\mathrm{GD}})
> > > = \mathbb{E}_{X,x_q,\epsilon}
> > > \Bigl[\bigl(x_q^\top(\tfrac{\eta}{C}X^\top X - I_d)\epsilon\bigr)^2\Bigr],
> > > $$
> > > which depends only on the centered variable $\epsilon = \hat w - w^\star \sim \mathcal N(0,I_d)$ and does not depend on the mean $w^\star$. In other words, shifting the prior mean $w^\star$ changes the individual GD iterates for each task, but leaves their expected squared error, and thus the dashed “one-step GD” baseline in Figure 5, unchanged. This is why the GD loss curves coincide for $w^\star = [1,\dots,1], [2,\dots,2],$ and $[3,\dots,3]$.
> > >
> > > We will integrate this answer in our final verison of manuscript.

---

> > > > ### Comment · Reviewer_2Dxp · 2025-11-19
> > > >
> > > > I would like to thank the authors for the further clarification, they have addressed my concerns.

---

### Review · Reviewer_tukB · 2025-10-17

**Summary Of Contributions:**

This paper investigates the connection between in-context learning (ICL) and gradient descent (GD) in a more realistic setting than prior work, specifically by relaxing the zero-initialization assumption commonly made in previous studies. The work primarily extends von Oswald et al.'s framework to non-zero initialization scenarios using linearized self-attention (LSA).
# Contributions:

- A theoretical proof demonstrating that when regression weights have a non-zero mean, multi-head LSA cannot generally replicate one-step GD, even with many heads. This is a significant theoretical result for understanding the fundamental limitations of LSA-based ICL.
- Theoretical upper bounds on the performance of LSA-based models based on number of heads.
- Empirical validation of the theoretical findings, including validation of prior mean w* impact.

# Strengths:

- Important theoretical contribution that advances understanding of when ICL can and cannot faithfully replicate gradient descent
- Generalizes existing work to more realistic settings (non-zero initialization)
- Empirical validation supports theoretical claims
- The mathematical proofs appear to be sound

# Weaknesses:

- Significant clarity and presentation issues that make the paper difficult to follow, especially for readers new to ICL research
- Critical missing information: dimensions of weight matrices (Wq, Wk, Wv, Wp) are not specified, making mathematical verification difficult.
- Insufficient explanation of linearized self-attention and its relationship to standard self-attention.
- Experimental setup is poorly explained (e.g., confusion about "training steps" in Figures 2 and 3 when ICL typically implies no training)
- Lack of intuition for key design choices (e.g., why multi-head outputs are summed rather than concatenated)
- Heavy reliance on external references to understand core concepts.

**Audience:**

Yes

**Audience Explanation:**

The paper addresses a fundamental question in the machine learning community: understanding when and how in-context learning relates to gradient descent. This is highly relevant given the widespread use of transformer-based models and growing interest in understanding their mechanisms.

**Broader Impact Concerns:**

This is primarily a theoretical paper investigating the mathematical relationship between in-context learning and gradient descent. The work does not raise obvious ethical concerns or require a Broader Impact Statement.

**Claims And Evidence:**

Yes

**Claims Explanation:**

The mathematical proofs appear to be sound, and the empirical evidence supports the theoretical claims, including validation of the prior mean w* impact. The core theoretical contribution regarding the impossibility of multi-head LSA replicating one-step GD with non-zero mean regression weights is convincing. However, the evidence is not presented clearly. The lack of dimensional specifications for key matrices makes independent verification challenging. The experimental setup lacks clarity, particularly regarding what “training” means in the context of ICL (Figures 2 and 3). While the claims appear to be supported, the presentation issues significantly hamper the ability to fully assess the evidence.

**Requested Changes:**

1.  Specify matrix dimensions explicitly: Provide clear dimensional specifications for all weight matrices (Wq, Wk, Wv, Wp) in the linear self-attention formulation. This is essential for readers to understand and verify the mathematical claims, particularly why updates are rank-one.
2. Clarify experimental setup:
Provide a clear, detailed explanation of the experimental methodology, particularly:
- What "training steps" mean in Figures 2 and 3 in the context of ICL (which typically involves no training)
- What exactly is being trained and how it relates to the ICL framework
- Make the experimental protocol self-contained and understandable for readers not deeply familiar with this specific line of work
3. Explain linearized self-attention thoroughly: Add a comprehensive explanation of linearized self-attention in the introduction or early sections:
Its relationship to standard self-attention ,Why and how it differs from traditional attention mechanisms ,Intuition for why it's an appropriate model for studying ICL-GD connections
4. Clarify multi-head aggregation design choice: Explain why multi-head attention outputs are summed (with each head having its own projection matrix) rather than using the standard concatenation approach in transformers. Provide motivation for this architectural choice.
5. Improve accessibility for ICL newcomers: The paper currently requires extensive familiarity with prior work. Add more intuitive explanations and self-contained descriptions of key concepts to make the paper accessible to a broader audience
6. Add intuition throughout: The paper would benefit from more intuitive explanations accompanying the mathematical formalism. For example, provide intuition for why the rank-one update structure emerges and why non-convexity of R(fH−LSA) is expected.
7. Improve figure captions: Make captions more informative and self-explanatory. For instance, "loss curves over the course of training" should specify what type of training and what is being optimized.

---

> ### Author Response · Authors · 2025-10-28
> **Author Response to Reviewer tukB**
>
> Thank you for the careful review and concrete suggestions. We respond point-by-point and will incorporate the requested clarifications in the revision.
>
> **Q1. Matrix dimensions and “rank-one” objects.**
>
> Let $E\in\mathbb{R}^{(d+1)\times(C+1)}$ be the embedding in Eqs. (2)–(3) of the paper, and $W_M\in\mathbb{R}^{(C+1)\times(C+1)}$ the mask. To make all products in Eq. (3) well-typed, we take
> $$
> W_Q,W_K,W_V,W_P\in\mathbb{R}^{(d+1)\times(d+1)}.
> $$
> In the proof of Theorem 1, the explicitly rank-one terms are the per-head contributions to
> $$
> M_i=\sum_{h=1}^H b_h,(a^{h}_i)^\top\ \in\ \mathbb{R}^{(d+1)\times(d+1)},\quad i\in[d{+}1],
> $$
> where each head contributes $b_h(a^{h}_i)^\top$ (outer product of two vectors), hence $\mathrm{rank}= 1$. This factorization is what drives the dimension-counting and the head-saturation result.
>
> **Q2. What is “training” in Figs. 2–3?**
>
> Our goal is to endow the LSA parameters with ICL capability in linear regression. Concretely, we train $\theta={W_Q,W_K,W_V,W_P}$ (and, for yq-LSA, also $w$) by minimizing the expected ICL risk
> $$
> \min_\theta\ \mathbb{E}\Big[(f_\theta(E)-y)^2\Big],
> $$
> where each Adam step samples fresh tasks $(X,y,x_q,y)$ from the task distribution and updates $\theta$. The x-axis “training steps” in Figs. 2–3 are these parameter-update iterations (no parameter updates occur at test time; ICL inference uses only the context). We will state this explicitly in Preliminaries and Experiments.
>
> **Q3. What is “linearized self-attention” here, and why use it?**
>
> Our LSA removes softmax and all nonlinearities (no LayerNorm/activation), yielding the form in Eq. (3). This choice (i) makes the update an affine function of simple context aggregates (inner products and cross-covariances), and (ii) enables closed-form analyses that isolate initialization effects. This follows the analytical tradition initiated by von Oswald et al. ([1] ICML2023) and extended in later work; non-linear/softmax settings that implement functional gradient methods are addressed, e.g., by Cheng–Chen–Sra ([2] ICML24).
>
> [1]Johannes von Oswald, Eyvind Niklasson, Ettore Randazzo, João Sacramento, Alexander Mordvintsev, Andrey Zhmoginov, and Max Vladymyrov. "Transformers learn in-context by gradient descent." Proceedings of Machine Learning Research
> [2]Xiang Cheng, Yuxin Chen, and Suvrit Sra. "Transformers Implement Functional Gradient Descent to Learn Non-Linear Functions In Context." In Forty-first International Conference on Machine Learning (ICML 2024).
>
> **Q4. Why sum multi-head outputs instead of concatenate?**
>
> Concatenation in standard Transformers is always followed by a linear projection $W_O$, algebraically this equals a sum of per-head linear maps. We fold that projection into per-head $W_P^{(h)}$ and sum:
> $$
> f_{\text{MH}}(E)=\Big[E + \sum_{h=1}^H \frac{1}{C} W_P^{(h)} W_V^{(h)} E \cdot W_M \big(E^\top (W_K^{(h)})^\top W_Q^{(h)} E\big)\Big]_{[-1,-1]}.
> $$
> This keeps the model dimension at $(d{+}1)$, is linearly equivalent to concat, and—crucially for Theorem 1—makes each head’s contribution an explicit rank-one update after reparameterization, which is what enables the clean head-saturation proof.
>
> **Q5–Q7. Accessibility, intuition, and figures.**
>
> We will (i) specify all matrix dimensions where first used; (ii) add an intuitive paragraph before each main result; (iii) expand captions to state precisely what is optimized (“parameter-training steps for expected ICL risk”), data generation, and baselines.
>
> We appreciate these suggestions, which substantially improve readability.
> All the above edits have been implemented in the revised manuscript and highlighted in green to mark where each clarification was added.

---

> > ### Comment · Reviewer_tukB · 2025-11-17
> > **Re:**
> >
> > Thanks to the authors for the clarifications, it addressed all my concerns.

---

### Review · Reviewer_3NnV · 2025-10-20

**Summary Of Contributions:**

This paper builds on the results of previous work that examined In-Context Learning (ICL) in simplified linear regression as a gradient descent step within the forward pass of a linear self-attention (LSA) mechanism. Earlier studies considered linear regression with a zero-mean Gaussian prior and constructed LSA weights such that the equivalence between ICL and gradient descent on linear regression would hold. Later, in Theorem 4, they showed that LSA behaves like gradient descent only if the query’s initial guess is a learnable parameter that matches the data’s prior mean.
Empirically, they first confirmed the effect of different initial guesses of the model in a toy example. In more realistic large language model (LLM) experiments, they used another model to provide a good initial guess in the initial prompt. Their results showed that a better initial guess leads to better in-context learning performance overall.

**Audience:**

Yes

**Audience Explanation:**

The results convincingly demonstrate that the deviation between ICL and GD stems not from limited expressivity but from a structural dependence on initialization. Theorem 1 shows that once the representational capacity of linear self-attention saturates at
H=d+1, adding more attention heads no longer brings it closer to gradient descent. This result matters because it establishes that the ICL–GD gap in this simplified setting is not an expressivity issue, but an inherent limitation of the mechanism itself.

Also they show that a zero-mean linear regression cannot be equivalent to the existing ICL of linear self-attention. However, The “no stationary point” impossibility of theorem 2 in the zero mean case is only for when the context goes to infinity C→∞.
 Then they show that an adaptive initialization can theoretically help with this mismatch. However, there are practical implications to a good initial guess:

- It implies the model can learn how to start thinking about each query, a kind of meta-initialization.

- It also changes the training problem: the model now must optimize not just transformations but also the inputs to its own inference.

This limits the applicability of the proposed framework empirically. However, the theoretical insights backed by valid experiments, gives a better understanding of ICL and is useful for the community.

**Claims And Evidence:**

Yes

**Claims Explanation:**

Yes. The claims are supported by clear theoretical arguments and empirical validation. The paper’s core proof demonstrates that even infinitely expressive LSA cannot emulate one-step GD under non-zero prior means, revealing a concrete limitation in prior equivalence claims. The accompanying experiments—both toy linear regression and LLM prompting—test and confirm this dependence on initialization, making the evidence both accurate and convincing.
My only concern is with the novelty of the theoretical results presented in this paper. I have asked for more clarification later.

**Requested Changes:**

My main concern is the novelty of the work given the current literature. I have take a look at: https://arxiv.org/pdf/2402.14951 after writing this review. The theoretical results sounds to be very similar. Can you state exactly what is the distinct theoretical contribution of yours? Also if a similar theorems exist in the literature, you must elaborate the exact distinction of your theorem with them in the main text. I don’t see this in your paper.

---

> ### Author Response · Authors · 2025-10-28
> **Author Response to Reviewer 3NnV**
>
> Thank you for the thoughtful review and for pointing us to Zhang et al. (arXiv:2402.14951). We agree that both papers study ICL in linear regression and both consider non-zero prior means. However, the questions asked and the resulting theorems are different and complementary:
>
> * **Research Goal.** Zhang et al. analyze a Linear Transformer Block (LTB) and show optimal LTB is equivalent to a one-step preconditioned GD (GD-β) and, in the large-context limit, approaches a (regularized) **Newton** step. Our work focuses on  LSA and asks when LSA itself can emulate one-step GD. More generally speaking, Zhang is discussing the role of MLP, while we are discussing the significance of guessing in the embedding input.
>
> * **Capacity Multihead LSA** We prove a head saturation result: the optimal ICL risk of multi-head LSA saturates at $H=d+1$. We also prove an impossibility (“no stationary point”) for multi-head LSA under zero-mean as. These establish that the ICL–GD gap is not due to insufficient expressivity beyond $d{+}1$ heads, but due to a structural dependence on initialization, a phenomenon not addressed in Zhang et al.
>
> * **Role of initialization.** Zhang et al. mitigate the gap by adding an MLP and let LTB realizing internally one-step Newton. We instead show that injecting the query initialization into the input—our yq-LSA with $y_q=w^\top x_q$—exactly recovers one-step GD under the appropriate prior, without any MLP. Thus, the two papers resolve the gap via different mechanisms (MLP vs. input-side initialization).
>
> In the revised manuscript, we have explicitly added subsections titled “Comparison with Concurrent Work” (highlighted in light blue), where we systematically contrast our theorems and assumptions with Zhang et al. and others work.
>
> We appreciate your suggestion—it strengthened our exposition and situates our contribution more clearly within the existing literature.

---

> > ### Comment · Reviewer_3NnV · 2025-11-19
> >
> > Thank you for the clarification and for updating the manuscript. Your response helped make the nuances of your contribution much clearer to me.

---

### Comment · Action_Editor_CA6J · 2025-12-05
**Missing emails in camera-ready**

Dear authors,

Thank you for submitting the camera-ready version with the requested revisions. I noticed that your emails are missing, please add them.

---

> ### Author Response · Authors · 2025-12-05
>
> Dear Action Editor,
>
> Thank you for the reminder. We have added the emails accordingly.

---

### Decision · Action_Editor_CA6J · 2025-11-24

**Recommendation:** Accept with minor revision

**Additional Comments:**

This paper studies the connection between in-context learning (ICL) and gradient descent (GD) in a more general setting than prior work. Existing results showed that ICL in the setting of linear regression corresponds to doing one-step GD in the forward pass of linear self-attention (LSA). These results assumed zero-mean Gaussian regression weights and zero initialization for GD. The authors study this connection with a non-zero mean $w_*$, showing that in this case multi-head LSA with infinite context does not replicate one-step GD, even with arbitrarily many heads. They introduce an extension of LSA with a trainable initialization $y_q = w^\top x_q$ of the new query's prediction. They show that this initialization plays an important role; LSA replicates one-step GD initialized at  $w_*$ if $w = w_*$ and again with infinite context. They empirically validate their theoretical results on a toy example. They also illustrated that adding a good initial guess, provided by another model, in the prompt of an LLM yield better in-context learning performance.

The results of this paper are closely related to the existing work of Zhang et al. (2024b), which also considered the non-zero mean setting. Zhang et al. showed that LSA alone will incur an irreducible additive approximation error in that setting. They also showed that a linear Transformer block (LTB), i.e., a combination of a linear multi-layer perceptron with an LSA, corresponds to one-step preconditioned GD with a learnable initialization, and that an optimal LTB with infinite context corresponds to one Newton step initialized at $w_*$.

Strengths:
- The problem studied is important and timely.
- The theoretical results presented advance the understanding of ICL and its connection to GD.
- Empirical results validate the theoretical results

Weaknesses:
- The analysis focuses on the infinite context setting.
- The results have limited practical applicability

All reviewers recommended to accept. The authors addressed in their responses all reviewers' concerns and edited the paper accordingly.

I am recommending to accept with the following minor revisions:

- Provide more motivation in the introduction on why it is important to consider the setting of non-zero mean.
- Discuss more clearly the connection to Zhang et al. (2024b) already in the introduction. Also define the acronym LTB first time you use it (currently you use it first on page 6 with no definition).
- Include the more detailed discussion of how the results change with finite C you provided in your response to Reviewer 2Dxp in the appendix of the paper.
- Add the following missing related work: Cheng, X., Chen, Y., & Sra, S. Transformers Implement Functional Gradient Descent to Learn Non-Linear Functions In Context. ICML 2024.

**Audience:**

Yes

**Audience Explanation:**

The paper contributes to a better understanding of ICL which is highly relevant to TMLR's audience.

**Claims And Evidence:**

Yes

**Claims Explanation:**

Theoretical claims are supported by proofs that appear sound and validated empirically.